# The nature of the last universal common ancestor and its impact on the early Earth system

Edmund R. R. Moody [1]✉, Sandra Álvarez-Carretero [1], Tara A. Mahendrarajah [2], James W. Clark[3], Holly C. Betts[1], Nina Dombrowski [2], Lénárd L. Szánthó [4,5,6], Richard A. Boyle[7], Stuart Daines[7], Xi Chen [8], Nick Lane [9], Ziheng Yang [9], Graham A. Shields [8], Gergely J. Szöllősi[5,6,10], Anja Spang [2,11], Davide Pisani [1,12]✉, Tom A. Williams [12]✉, Timothy M. Lenton [7]✉ & Philip C. J. Donoghue [1]✉

The nature of the last universal common ancestor (LUCA), its age and its impact on the Earth system have been the subject of vigorous debate across diverse disciplines, often based on disparate data and methods. Age estimates for LUCA are usually based on the fossil record, varying with every reinterpretation. The nature of LUCA's metabolism has proven equally contentious, with some attributing all core metabolisms to LUCA, whereas others reconstruct a simpler life form dependent on geochemistry. Here we infer that LUCA lived ~4.2 Ga (4.09–4.33 Ga) through divergence time analysis of pre-LUCA gene duplicates, calibrated using microbial fossils and isotope records under a new cross-bracing implementation. Phylogenetic reconciliation suggests that LUCA had a genome of at least 2.5 Mb (2.49–2.99 Mb), encoding around 2,600 proteins, comparable to modern prokaryotes. Our results suggest LUCA was a prokaryote-grade anaerobic acetogen that possessed an early immune system. Although LUCA is sometimes perceived as living in isolation, we infer LUCA to have been part of an established ecological system. The metabolism of LUCA would have provided a niche for other microbial community members and hydrogen recycling by atmospheric photochemistry could have supported a modestly productive early ecosystem.

The common ancestry of all extant cellular life is evidenced by the universal genetic code, machinery for protein synthesis, shared chirality of the almost-universal set of 20 amino acids and use of ATP as a common energy currency[1]. The last universal common ancestor (LUCA) is the node on the tree of life from which the fundamental prokaryotic domains (Archaea and Bacteria) diverge. As such, our understanding of LUCA impacts our understanding of the early evolution of life on Earth. Was LUCA a simple or complex organism? What kind of environment did

it inhabit and when? Previous estimates of LUCA are in conflict either due to conceptual disagreement about what LUCA is[2] or as a result of different methodological approaches and data[3–9]. Published analyses differ in their inferences of LUCA's genome, from conservative estimates of 80 orthologous proteins[10] up to 1,529 different potential gene families[4]. Interpretations range from little beyond an information-processing and metabolic core[6] through to a prokaryote-grade organism with much of the gene repertoire of modern Archaea and Bacteria[8], recently

A full list of affiliations appears at the end of the paper. ✉e-mail: edmund.moody@bristol.ac.uk; davide.pisani@bristol.ac.uk; tom.a.williams@bristol.ac.uk; T.M.Lenton@exeter.ac.uk; phil.donoghue@bristol.ac.uk

reviewed in ref. 7. Here we use molecular clock methodology, horizontal gene-transfer-aware phylogenetic reconciliation and existing biogeochemical models to address questions about LUCA's age, gene content, metabolism and impact on the early Earth system.

## Estimating the age of LUCA

Life's evolutionary timescale is typically calibrated to the oldest fossil occurrences. However, the veracity of fossil discoveries from the early Archaean period has been contested[11,12]. Relaxed Bayesian node-calibrated molecular clock approaches provide a means of integrating the sparse fossil and geochemical record of early life with the information provided by molecular data; however, constraining LUCA's age is challenging due to limited prokaryote fossil calibrations and the uncertainty in their placement on the phylogeny. Molecular clock estimates of LUCA[13–15] have relied on conserved universal single-copy marker genes within phylogenies for which LUCA represented the root. Dating the root of a tree is difficult because errors propagate from the tips to the root of the dated phylogeny and information is not available to estimate the rate of evolution for the branch incident on the root node. Therefore, we analysed genes that duplicated before LUCA with two (or more) copies in LUCA's genome[16]. The root in these gene trees represents this duplication preceding LUCA, whereas LUCA is represented by two descendant nodes. Use of these universal paralogues also has the advantage that the same calibrations can be applied at least twice. After duplication, the same species divergences are represented on both sides of the gene tree[17,18] and thus can be assumed to have the same age. This considerably reduces the uncertainty when genetic distance (branch length) is resolved into absolute time and rate. When a shared node is assigned a fossil calibration, such cross-bracing also serves to double the number of calibrations on the phylogeny, improving divergence time estimates. We calibrated our molecular clock analyses using 13 calibrations (see 'Fossil calibrations' in Supplementary Information). The calibration on the root of the tree of life is of particular importance. Some previous studies have placed a younger maximum constraint on the age of LUCA based on the assumption that life could not have survived Late Heavy Bombardment (LHB) (-3.7–3.9 billion years ago (Ga))[19]. However, the LHB hypothesis is extrapolated and scaled from the Moon's impact record, the interpretation of which has been questioned in terms of the intensity, duration and even the veracity of an LHB episode[20–23]. Thus, the LHB hypothesis should not be considered a credible maximum constraint on the age of LUCA. We used soft-uniform bounds, with the maximum-age bound based on the time of the Moon-forming impact (4,510 million years ago (Ma) ± 10 Myr), which would have effectively sterilized Earth's precursors, Tellus and Theia[13]. Our minimum bound on the age of LUCA is based on low $\delta^{98}Mo$ isotope values indicative of Mn oxidation compatible with oxygenic photosynthesis and, therefore, total-group Oxyphotobacteria in the Mozaan Group, Pongola Supergroup, South Africa[24,25], dated minimally to 2,954 Ma ± 9 Myr (ref. 26).

Our estimates for the age of LUCA are inferred with a concatenated and a partitioned dataset, both consisting of five pre-LUCA paralogues: catalytic and non-catalytic subunits from ATP synthases, elongation factor Tu and G, signal recognition protein and signal recognition particle receptor, tyrosyl-tRNA and tryptophanyl-tRNA synthetases, and leucyl- and valyl-tRNA synthetases[27]. Marginal densities (commonly referred to as effective priors) fall within calibration densities (that is, user-specified priors) when topologically adjacent calibrations do not overlap temporally, but may differ when they overlap, to ensure the relative age relationships between ancestor-descendant nodes. We consider the marginal densities a reasonable interpretation of the calibration evidence given the phylogeny; we are not attempting to test the hypothesis that the fossil record is an accurate temporal archive of evolutionary history because it is not[28]. The duplicated LUCA node age estimates we obtained under the autocorrelated rates (geometric Brownian motion (GBM))[29,30] and independent-rates log-normal

(ILN)[31,32] relaxed-clock models with our partitioned dataset (GBM, 4.18–4.33 Ga; ILN, 4.09–4.32 Ga; Fig. 1) fall within our composite age estimate for LUCA ranging from 3.94 Ga to 4.52 Ga, comparable to previous studies[13,18,33]. Dating analyses based on single genes, or concatenations that excluded each gene in turn, returned compatible timescales (Extended Data Figs. 1 and 2 and 'Additional methods' in Methods).

## LUCA's physiology

To estimate the physiology of LUCA, we first inferred an updated microbial phylogeny from 57 phylogenetic marker genes (see 'Universal marker genes' in Methods) on 700 genomes, comprising 350 Archaea and 350 Bacteria[15]. This tree was in good agreement with recent phylogenies of the archaeal and bacterial domains of life[34,35]. For example, the TACK[36] and Asgard clades of Archaea[37–39] and Gracilicutes within Bacteria[40,41] were recovered as monophyletic. However, the analysis was equivocal as to the phylogenetic placement of the Patescibacteria (CPR)[42] and DPANN[43], which are two small-genome lineages that have been difficult to place in trees. Approximately unbiased[44] tests could not distinguish the placement of these clades, neither at the root of their respective domains nor in derived positions, with CPR sister to Chloroflexota (as reported recently in refs. 35,41,45) and DPANN sister to Euryarchaeota. To account for this phylogenetic uncertainty, we performed LUCA reconstructions on two trees: our maximum likelihood (ML) tree (topology 1; Extended Data Fig. 3) and a tree in which CPR were placed as the sister of Chloroflexota, with DPANN sister to all other Archaea (topology 2; Extended Data Fig. 4). In both cases, the gene families mapped to LUCA were very similar (correlation of LUCA presence probabilities (PP), $r = 0.6720275$, $P < 2.2 \times 10^{-16}$). We discuss the results on the tree with topology 2 and discuss the residual differences in Supplementary Information, 'Topology 1' (Supplementary Data 1).

We used the probabilistic gene- and species-tree reconciliation algorithm ALE[46] to infer the evolution of gene family trees for each sampled entry in the KEGG Orthology (KO) database[47] on our species tree. ALE infers the history of gene duplications, transfers and losses based on a comparison between a distribution of bootstrapped gene trees and the reference species tree, allowing us to estimate the probability that the gene family was present at a node in the tree[35,48,49]. This reconciliation approach has several advantages for drawing inferences about LUCA. Most gene families have experienced gene transfer since the time of LUCA[50,51] and so explicitly modelling transfers enables us to include many more gene families in the analysis than has been possible using previous approaches. As the analysis is probabilistic, we can also account for uncertainty in gene family origins and evolutionary history by averaging over different scenarios using the reconciliation model. Using this approach, we estimated the probability that each KEGG gene family (KO) was present in LUCA and then used the resulting probabilities to construct a hypothetical model of LUCA's gene content, metabolic potential (Fig. 2) and environmental context (Fig. 3). Using the KEGG annotation is beneficial because it allows us to connect our inferences to curated functional annotations; however, it has the drawback that some widespread gene families that were likely present in LUCA are divided into multiple KO families that individually appear to be restricted to particular taxonomic groups and inferred to have arisen later. To account for this limitation, we also performed an analysis of COG (Clusters of Orthologous Genes)[52] gene families, which correspond to more coarse-grained functional annotations (Supplementary Data 2).

### Genome size and cellular features

By using modern prokaryotic genomes as training data, we used a predictive model to estimate the genome size and the number of protein families encoded by LUCA based on the relationship between the number of KEGG gene families and the total number of proteins encoded by modern prokaryote genomes (Extended Data Figs. 5 and 6). On the basis of the PPs for KEGG KO gene families, we identified a conservative

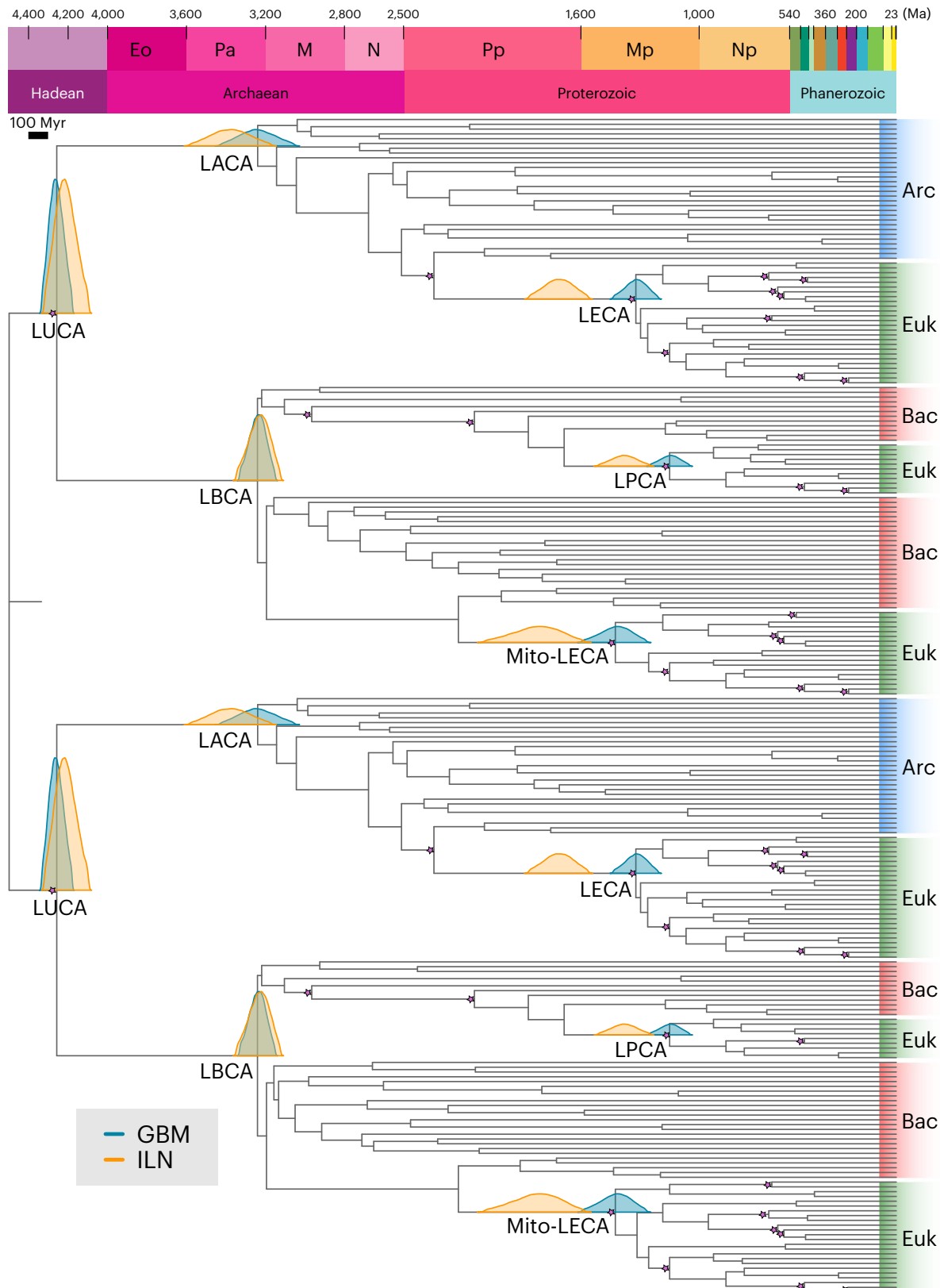

**Fig. 1 | Timetree inferred under a Bayesian node-dating approach with cross-bracing using a partitioned dataset of five pre-LUCA paralogues.** Our results suggest that LUCA lived around 4.2 Ga, with a 95% confidence interval spanning 4.09–4.33 Ga under the ILN relaxed-clock model (orange) and 4.18–4.33 Ga under the GBM relaxed-clock model (teal). Under a cross-bracing approach, nodes corresponding to the same species divergences (that is, mirrored nodes) have the same posterior time densities. This figure shows the corresponding posterior time densities of the mirrored nodes for the last universal, archaeal, bacterial and eukaryotic common ancestors (LUCA, LACA, LBCA and LECA, respectively); the last common ancestor of the mitochondrial lineage (Mito-LECA); and the last plastid-bearing common ancestor (LPCA). Purple stars indicate nodes calibrated with fossils. Arc, Archaea; Bac, Bacteria; Euk, Eukarya.

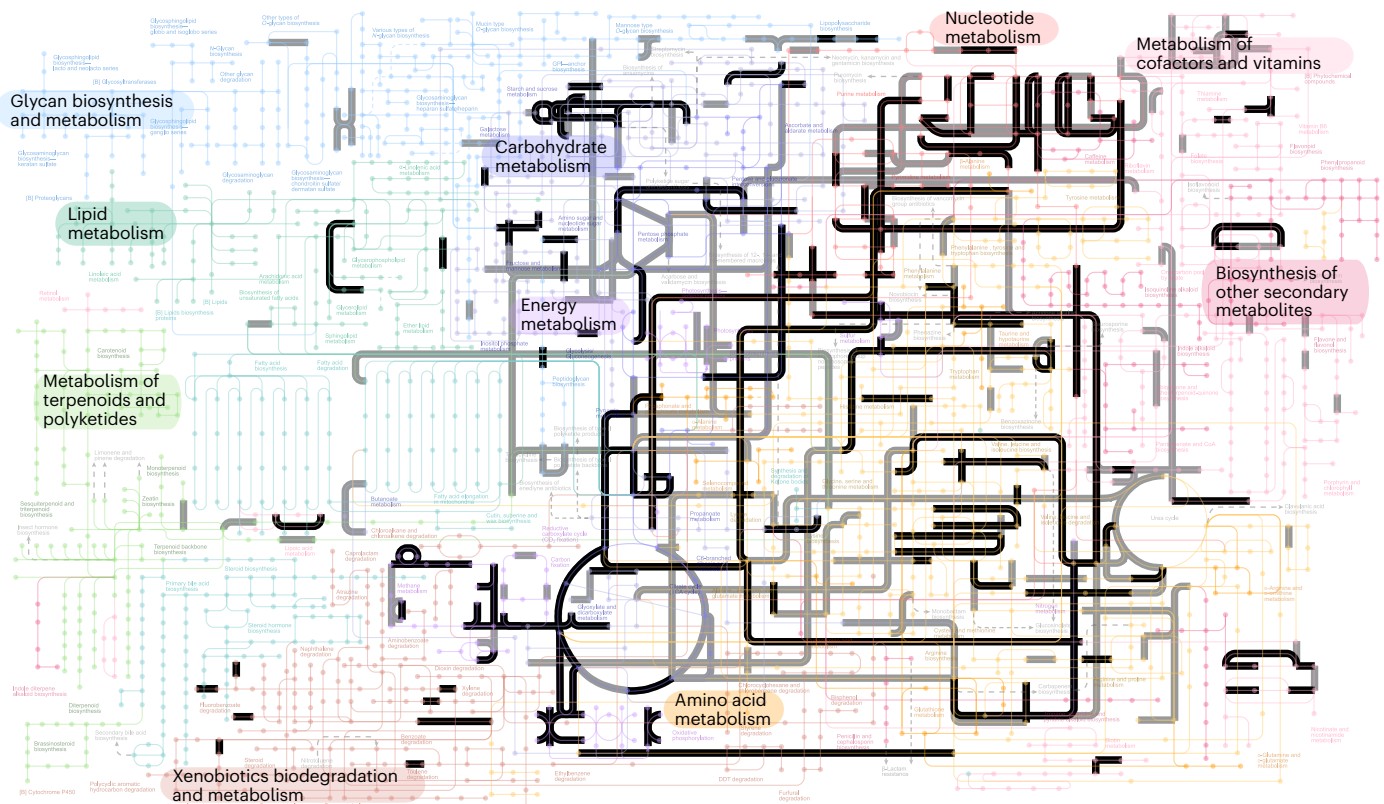

**Fig. 2 | Probabilistic estimates of metabolic networks from modern life that were present in LUCA.** In black: enzymes and metabolic pathways inferred to be present in LUCA with at least PP = 0.75, with sampling in both prokaryotic domains. In grey: those inferred in our least-stringent threshold of PP = 0.50.

The analysis supports the presence of a complete WLP and an almost complete TCA cycle across multiple confidence thresholds. Metabolic maps derived from KEGG[47] database through iPath[109]. GPI, glycosylphosphatidylinositol; DDT, 1,1,1-trichloro-2,2-bis(p-chlorophenyl)ethane .

subset of 399 KOs that were likely to be present in LUCA, with PPs ≥0.75, and found in both Archaea and Bacteria (Supplementary Data 1); these families form the basis of our metabolic reconstruction. However, by integrating over the inferred PPs of all KO gene families, including those with low probabilities, we also estimate LUCA's genome size. Our predictive model estimates a genome size of 2.75 Mb (2.49–2.99 Mb) encoding 2,657 (2,451–2,855) proteins (Methods). Although we can estimate the number of genes in LUCA's genome, it is more difficult to identify the specific gene families that might have already been present in LUCA based on the genomes of modern Archaea and Bacteria. It is likely that the modern version of the pathways would be considered incomplete based on LUCA's gene content through subsequent evolutionary changes. We should therefore expect reconstructions of metabolic pathways to be incomplete due to this phylogenetic noise and other limitations of the analysis pipeline. For example, when looking at genes and pathways that can uncontroversially be mapped to LUCA, such as the ribosome and aminoacyl-tRNA synthetases for implementing the genetic code, we find that we map many (but not all) of the key components to LUCA (see 'Notes' in Supplementary Information). We interpret this to mean that our reconstruction is probably incomplete but our interpretation of LUCA's metabolism relies on our inference of pathways, not individual genes.

The inferred gene content of LUCA suggests it was an anaerobe as we do not find support for the presence of terminal oxidases (Supplementary Data 1). Instead we identified almost all genes encoding proteins of the archaeal (and most of the bacterial) versions of the Wood–Ljungdahl pathway (WLP) (PP > 0.7), indicating that LUCA had the potential for acetogenic growth and/or carbon fixation[53–55] (Supplementary Data 3). LUCA encoded some NiFe hydrogenase subunits (K06281, PP = 0.90; K14126, PP = 0.92), which may have enabled growth

on hydrogen (see 'Notes' in Supplementary Information). Complexes involved in methanogenesis such as methyl-coenzyme M reductase and tetrahydromethanopterin S-methyltransferase were inferred to be absent, suggesting that LUCA was unlikely to function as a modern methanogen. We found strong support for some components of the TCA cycle (including subunits of oxoglutarate/2-oxoacid ferredoxin oxidoreductase (K00175 and K00176), succinate dehydrogenase (K00239) and homocitrate synthase (K02594)), although some steps are missing. LUCA was probably capable of gluconeogenesis/glycolysis in that we find support for most subunits of enzymes involved in these pathways (Supplementary Data 1 and 3). Considering the presence of the WLP, this may indicate that LUCA had the ability to grow organoheterotrophically and potentially also autotrophically. Gluconeogenesis would have been important in linking carbon fixation to nucleotide biosynthesis via the pentose phosphate pathway, most enzymes of which seem to be present in LUCA (see 'Notes' in Supplementary Information). We found no evidence that LUCA was photosynthetic, with low PPs for almost all components of oxygenic and anoxygenic photosystems (Supplementary Data 3).

We find strong support for the presence of ATP synthase, specifically, the A (K02117, PP = 0.98) and B (K02118, PP = 0.94) subunit components of the hydrophilic V/A1 subunit, and the I (subunit a, K02123, PP = 0.99) and K (subunit c, K02124, PP = 0.82) subunits of the transmembrane V/A0 subunit. In addition, if we relax the sampling threshold, we also infer the presence of the F1-type β-subunit (K02112, PP = 0.94). This is consistent with many previous studies that have mapped ATP synthase subunits to LUCA[6,17,18,56,57].

We obtain moderate support for the presence of pathways for assimilatory nitrate (ferredoxin-nitrate reductase, K00367, PP = 0.69; ferredoxin-nitrite reductase, K00367, PP = 0.53) and sulfate reduction

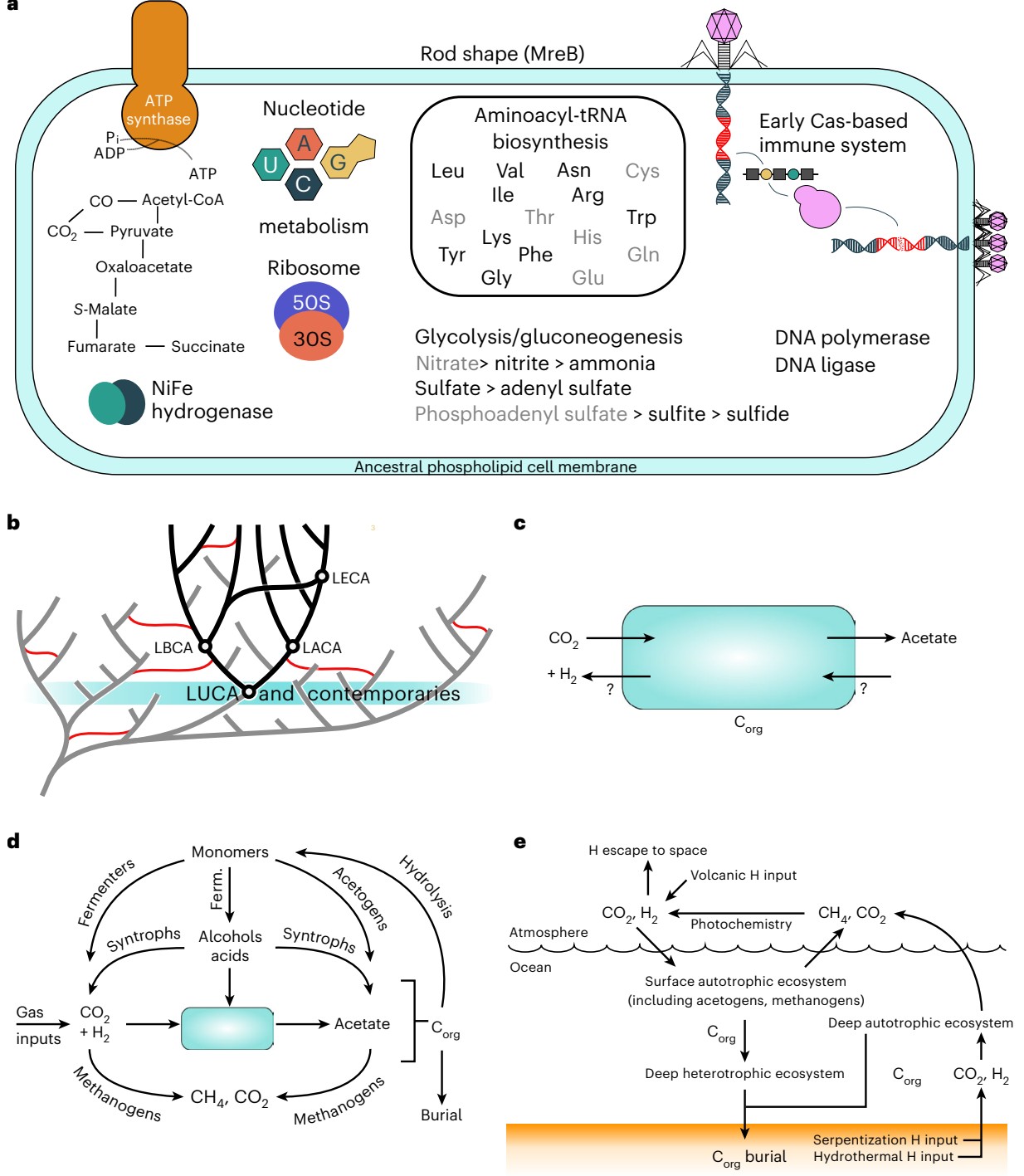

**Fig. 3 | A reconstruction of LUCA, within its evolutionary and ecological context. a**, A representation of LUCA based on our ancestral gene content reconstruction. Gene names in black have been inferred to be present in LUCA under the most-stringent threshold (PP = 0.75, sampled in both domains); those in grey are present at the least-stringent threshold (PP = 0.50, without a requirement for presence in both domains). **b**, LUCA in the context of the tree of life. Branches on the tree of life that have left sampled descendants today are coloured black, those that have left no sampled descendants are in grey. As the common ancestor of extant cellular life, LUCA is the oldest node that can be reconstructed using phylogenetic methods. It would have shared the early Earth with other lineages (highlighted in teal) that have left no descendants among sampled cellular life today. However, these lineages may have left a trace in modern organisms by transferring genes into the sampled tree of life

(red lines) before their extinction. **c**, LUCA's chemoautotrophic metabolism probably relied on gas exchange with the immediate environment to achieve organic carbon ($C_{org}$) fixation via acetogenesis and it may also have run the metabolism in reverse. **d**, LUCA within the context of an early ecosystem. The $CO_2$ and $H_2$ that fuelled LUCA's plausibly acetogenic metabolism could have come from both geochemical and biotic inputs. The organic matter and acetate that LUCA produced could have created a niche for other metabolisms, including ones that recycled $CO_2$ and $H_2$ (as in modern sediments). **e**, LUCA in an Earth system context. Acetogenic LUCA could have been a key part of both surface and deep (chemo)autotrophic ecosystems, powered by $H_2$. If methanogens were also present, hydrogen would be released as $CH_4$ to the atmosphere, converted to $H_2$ by photochemistry and thus recycled back to the surface ecosystem, boosting its productivity. Ferm., fermentation.

(sulfate adenylyltransferase, K00957, PP = 0.80, and K00958, PP = 0.73; sulfite reductase, K00392, PP = 0.82; phosphoadenosine phosphosulfate reductase, K00390, PP = 0.56), probably to fuel amino acid biosynthesis, for which we inferred the presence of 37 partially complete pathways.

We found support for the presence of 19 class 1 CRISPR–Cas effector protein families in the genome of LUCA, including types I and III (cas3, K07012, PP = 0.80, and K07475, PP = 0.74; cas10, K07016, PP = 0.96, and K19076, PP = 0.67; and cas7, K07061, PP = 0.90, K09002, PP = 0.84, K19075, PP = 0.97, K19115, PP = 0.98, and K19140, PP = 0.80). The absence of Cas1 and Cas2 may suggest LUCA encoded an early Cas system with the means to deliver an RNA-based immune response by cutting (Cas6/Cas3) and binding (CSM/Cas10) RNA, but lacking the full immune-system-site CRISPR. This supports the idea that the effector stage of CRISPR–Cas immunity evolved from RNA sensing for signal transduction, based on the similarities in RNA binding modules of the proteins[58]. This is consistent with the idea that cellular life was already involved in an arms race with viruses at the time of LUCA[59,60]. Our results indicate that an early Cas system was an ancestral immune system of extant cellular life.

Altogether, our metabolic reconstructions suggest that LUCA was a relatively complex organism, similar to extant Archaea and Bacteria[6,7]. On the basis of ancient duplications of the *Sec* and ATP synthase genes before LUCA, along with high PPs for key components of those systems, membrane-bound ATP synthase subunits, genes involved in peptidoglycan synthesis (*mraY*, K01000; *murC*, K01924) and the cytoskeletal actin-like protein, MreB (K03569) (Supplementary Data 3), it is highly likely that LUCA possessed the core cellular apparatus of modern prokaryotic life. This might include the basic constituents of a phospholipid membrane, although our analysis did not conclusively establish its composition. In particular, we recovered the following enzymes involved in the synthesis of ether and ester lipids, (alkyldihydroxyacetonephosphate synthase, glycerol 3-phosphate and glycerol 1-phosphate) and components of the mevalonate pathway (mevalonate 5-phosphate dehydratase (PP = 0.84), hydroxymethylglutaryl-CoA reductase (PP = 0.52), mevalonate kinase (PP = 0.51) and hydroxymethylglutaryl-CoA synthase (PP = 0.51)).

Compared with previous estimates of LUCA's gene content, we find 81 overlapping COG gene families with the consensus dataset of ref. 7 and 69 overlapping KOs with the dataset of ref. 6. Key points of agreement between previous studies include the presence of signal recognition particle protein, ffh (COG0541, K03106)[7] used in the targeting and delivery of proteins for the plasma membrane, a high number of aminoacyl-tRNA synthetases for amino acid synthesis and glycolysis/gluconeogenesis enzymes.

Ref. 6 inferred LUCA to be a thermophilic anaerobic autotroph using the WLP for carbon fixation based on the presence of a single enzyme (CODH), and similarly suggested that LUCA was capable of nitrogen fixation using a nitrogenase. Our reconstruction agrees with ref. 6 that LUCA was an anaerobic autotroph using the WLP for carbon fixation, but we infer the presence of a much more complete WLP than that previously obtained. We did not find strong evidence for nitrogenase or nitrogen fixation, and the reconstruction was not definitive with respect to the optimal growth environment of LUCA.

We used a probabilistic approach to reconstruct LUCA—that is, we estimated the probability with which each gene family was present in LUCA based on a model of how gene families evolve along an overarching species tree. This approach differs from analyses of phylogenetic presence–absence profiles[3,4,9] or those that used filtering criteria (such as broadly distributed or highly vertically evolving families) to define a high-confidence subset of modern genes that might have been present in LUCA. Our reconstruction maps many more genes to LUCA—albeit each with lower probability—than previous analyses[8] and yields an estimate of LUCA's genome size that is within the range of modern prokaryotes. The result is an incomplete picture of a cellular organism that was prokaryote

grade rather than progenotic[2] and that, similarly to prokaryotes today, probably existed as part of an ecosystem. As the common ancestor of sampled, extant prokaryotic life, LUCA is the oldest node on the species tree that we can reconstruct via phylogenomics but, as Fig. 3 illustrates, it was already the product of a highly innovative period in evolutionary history during which most of the core components of cells were established. By definition, we cannot reconstruct LUCA's contemporaries using phylogenomics but we can propose hypotheses about their physiologies based on the reconstructed LUCA whose features immediately suggest the potential for interactions with other prokaryotic metabolisms.

## LUCA's environment, ecosystem and Earth system context

The inference that LUCA used the WLP helps constrain the environment and ecology in which it could have lived. Modern acetogens can grow autotrophically on $H_2$ (and $CO_2$) or heterotrophically on a wide range of alternative electron donors including alcohols, sugars and carboxylic acids[55]. This metabolic flexibility is key to their modern ecological success. Acetogenesis, whether autotrophic or heterotrophic, has a low energy yield and growth efficiency (although use of the reductive acetyl-CoA pathway for both energy production and biosynthesis reduces the energy cost of biosynthesis). This would be consistent with an energy-limited early biosphere[61].

If LUCA functioned as an organoheterotrophic acetogen, it was necessarily part of an ecosystem containing autotrophs providing a source of organic compounds (because the abiotic source flux of organic molecules was minimal on the early Earth). Alternatively, if LUCA functioned as a chemoautotrophic acetogen it could (in principle) have lived independently off an abiotic source of $H_2$ (and $CO_2$). However, it is implausible that LUCA would have existed in isolation as the by-products of its chemoautotrophic metabolism would have created a niche for a consortium of other metabolisms (as in modern sediments) (Fig. 3d). This would include the potential for LUCA itself to grow as an organoheterotroph.

A chemoautotrophic acetogenic LUCA could have occupied two major potential habitats (Fig. 3e): the first is the deep ocean where hydrothermal vents and serpentinization of sea-floor provided a source of $H_2$ (ref. 62). Consistent with this, we find support for the presence of reverse gyrase (PP = 0.97), a hallmark enzyme of hyperthermophilic prokaryotes[6,63–65], which would not be expected if early life existed at the ocean surface (although the evolution of reverse gyrase is complex[63]; see 'Reverse gyrase' in Supplementary Information). The second habitat is the ocean surface where the atmosphere would have provided a source of $H_2$ derived from volcanoes and metamorphism. Indeed, we detected the presence of spore photoproduct lyase (COG1533, K03716, PP = 0.88) that in extant organisms repairs methylene-bridged thymine dimers occurring in spore DNA as a result of damage induced through ultraviolet (UV) radiation[66,67]. However, this gene family also occurs in modern taxa that neither form endospores nor dwell in environments where they are likely to accrue UV damage to their DNA and so is not an exclusive hallmark of environments exposed to UV. Previous studies often favoured a deep-ocean environment for LUCA as early life would have been better protected there from an episode of LHB. However, if the LHB was less intense than initially proposed[20,22], or just a sampling artefact[21], these arguments weaken. Another possibility may be that LUCA inhabited a shallow hydrothermal vent or a hot spring.

Hydrogen fluxes in these ecosystems could have been several times higher on the early Earth (with its greater internal heat source) than today. Volcanism today produces ~$1 × 10^{12}$ mol $H_2$ yr$^{-1}$ and serpentinization produces ~$0.4 × 10^{12}$ mol $H_2$ yr$^{-1}$. With the present $H_2$ flux and the known scaling of the $H_2$ escape rate to space, an abiotic atmospheric concentration of $H_2$ of ~150 ppmv is predicted[68]. Chemoautotrophic acetogens would have locally drawn down the concentration of $H_2$ (in either surface or deep niche) but their low growth efficiency would ensure $H_2$ (and $CO_2$) remained available. This and the organic matter and

acetate produced would have created niches for other metabolisms, including methanogenesis (Fig. 3d).

On the basis of thermodynamic considerations, $CH_4$ and $CO_2$ are expected to be the eventual metabolic end products of the resulting ecosystem, with a small fraction of the initial hydrogen consumption buried as organic matter. The resulting flux of $CH_4$ to the atmosphere would fuel photochemical $H_2$ regeneration and associated productivity in the surface ocean (Fig. 3e). Existing models suggest the resulting global $H_2$ recycling system is highly effective, such that the supply flux of $H_2$ to the surface could have exceeded the volcanic input of $H_2$ to the atmosphere by at least an order of magnitude, in turn implying that the productivity of such a biosphere was boosted by a comparable factor[69]. Photochemical recycling to CO would also have supported a surface niche for organisms consuming CO (ref. 69).

In deep-ocean habitats, there could be some localized recycling of electrons (Fig. 3d) but a quantitative loss of highly insoluble $H_2$ and $CH_4$ to the atmosphere and minimal return after photochemical conversion of $CH_4$ to $H_2$ means global recycling to depth would be minimal (Fig. 3e). Hence the surface environment for LUCA could have become dominant (albeit recycling of the resulting organic matter could be spread through ocean depth; 'Deep heterotrophic ecosystem' in Fig. 3e). The global net primary productivity of an early chemoautotrophic biosphere including acetogenic LUCA and methanogens could have been of order ~$1 \times 10^{12}$ to $7 \times 10^{12}$ mol C yr$^{-1}$ (~3 orders of magnitude less than today)[69].

The nutrient supply (for example, N) required to support such a biosphere would need to balance that lost in the burial flux of organic matter. Earth surface redox balance dictates that hydrogen loss to space and burial of electrons/hydrogen must together balance input of electrons/hydrogen. Considering contemporary $H_2$ inputs, and the above estimate of net primary productivity, this suggests a maximum burial flux in the order of ~$10^{12}$ mol C yr$^{-1}$, which, with contemporary stoichiometry (C:N ratio of ~7) could demand >$10^{11}$ mol N yr$^{-1}$. Lightning would have provided a source of nitrite and nitrate[70], consistent with LUCA's inferred pathways of nitrite and (possibly) nitrate reduction. However, it would only have been of the order $3 \times 10^9$ mol N yr$^{-1}$ (ref. 71). Instead, in a global hydrogen-recycling system, HCN from photochemistry higher in the atmosphere, deposited and hydrolysed to ammonia in water, would have increased available nitrogen supply by orders of magnitude toward ~$3 \times 10^{12}$ mol N yr$^{-1}$ (refs. 71,72). This HCN pathway is consistent with the anomalously light nitrogen isotopic composition of the earliest plausible biogenic matter of 3.8–3.7 Ga (ref. 73), although that considerably postdates our inferred age of LUCA. These considerations suggest that the proposed LUCA biosphere (Fig. 3e) would have been energy or hydrogen limited not nitrogen limited.

## Conclusions

By treating gene presence probabilistically, our reconstruction maps many more genes (2,657) to LUCA than previous analyses and results in an estimate of LUCA's genome size (2.75 Mb) that is within the range of modern prokaryotes. The result is a picture of a cellular organism that was prokaryote grade rather than progenotic[2] and that probably existed as a component of an ecosystem, using the WLP for acetogenic growth and carbon fixation. We cannot use phylogenetics to reconstruct other members of this early ecosystem but we can infer their physiologies based on the metabolic inputs and outputs of LUCA. How evolution proceeded from the origin of life to early communities at the time of LUCA remains an open question, but the inferred age of LUCA (~4.2 Ga) compared with the origin of the Earth and Moon suggests that the process required a surprisingly short interval of geologic time.

## Methods

### Universal marker genes

A list of 298 markers were identified by creating a non-redundant list of markers used in previous studies on archaeal and bacterial phylogenies[10,35,38,74–79]. These markers were mapped to the corresponding COG, arCOG and TIGRFAM profile to identify which profile is best suited to extract proteins from taxa of interest. To evaluate whether the markers cover all archaeal and bacterial diversity, proteins from a set of 574 archaeal and 3,020 bacterial genomes were searched against the COG, arCOG and TIGRFAM databases using hmmsearch (v.3.1b2; settings, hmmsearch–tblout output–domtblout–notextw)[52,80–82]. Only hits with an e-value less than or equal to $1 \times 10^{-5}$ were investigated further and for each protein the best hit was determined based on the e-value (expect value) and bit-score. Results from all database searches were merged based on the protein identifiers and the table was subsetted to only include hits against the 298 markers of interest. On the basis of this table we calculated whether the markers occurred in Archaea, Bacteria or both Archaea and Bacteria. Markers were only included if they were present in at least 50% of taxa and contained less than 10% of duplications, leaving a set of 265 markers. Sequences for each marker were aligned using MAFFT L-INS-i v.7.407 (ref. 83) for markers with less than 1,000 sequences or MAFFT[84] for those with more than 1,000 sequences (setting, –reorder)[84] and sequences were trimmed using BMGE[85], set for amino acids, a BLOcks SUbstitution Matrix 30 similarity matrix, with a entropy score of 0.5 (v.1.12; settings, -t AA -m BLOSUM30 -h 0.5). Single gene trees were generated with IQ-TREE 2 (ref. 86), using the LG substitution matrix, with ten-profile mixture models, four CPUs, with 1,000 ultrafast bootstraps optimized by nearest neighbour interchange written to a file retaining branch lengths (v.2.1.2; settings, -m LG + C10 + F + R -nt 4 -wbtl -bb 1,000 -bnni). These single gene trees were investigated for archaeal and bacterial monophyly and the presence of paralogues. Markers that failed these tests were not included in further analyses, leaving a set of 59 markers (3 arCOGs, 46 COGs and 10 TIGRFAMs) suited for phylogenies containing both Archaea and Bacteria (Supplementary Data 4).

### Marker gene sequence selection

To limit selecting distant paralogues and false positives, we used a bidirectional or reciprocal approach to identify the sequences corresponding to the 59 single-copy markers. In the first inspection (query 1), the 350 archaeal and 350 bacterial reference genomes were queried against all arCOG HMM (hidden Markov model) profiles (All_Arcogs_2018.hmm), all COG HMM profiles (NCBI_COGs_Oct2020.hmm) and all TIGRFAM HMM profiles (TIGRFAMs_15.0_HMM.LIB) using a custom script built on hmmsearch: hmmsearchTable <genomes.faa> <database.hmm> -E 1 × $10^{-5}$ >HMMscan_Output_e5 (HMMER v.3.3.2)[87]. HMM profiles corresponding to the 59 single-copy marker genes (Supplementary Data 4) were extracted from each query and the best-hit sequences were identified based on the e-value and bit-score. We used the same custom hmmsearchTable script and conditions (see above) in the second inspection (query 2) to query the best-hit sequences identified above against the full COG HMM database (NCBI_COGs_Oct2020.hmm). Results were parsed and the COG family assigned in query 2 was compared with the COG family assigned to sequences based on the marker gene identity (Supplementary Data 4). Sequence hits were validated using the matching COG identifier, resulting in 353 mismatches (that is, COG family in query 1 does not match COG family in query 2) that were removed from the working set of marker gene sequences. These sequences were aligned using MAFFT L-INS-i[83] and then trimmed using BMGE[85] with a BLOSUM30 matrix. Individual gene trees were inferred under ML using IQ-TREE 2 (ref. 86) with model fitting, including both the default homologous substitution models and the following complex heterogeneous substitution models (LG substitution matrices with 10–60-profile mixture models, with empirical base frequencies and a discrete gamma model with four categories accounting for rate heterogeneity across sites): LG + C60 + F + G, LG + C50 + F + G, LG + C40 + F + G, LG + C30 + F + G, LG + C20 + F + G and LG + C10 + F + G, with 10,000 ultrafast bootstraps and 10 independent runs to avoid local optima. These 59 gene trees were manually inspected and curated over

multiple rounds. Any horizontal gene transfer events, paralogous genes or sequences that violated domain monophyly were removed and two genes (arCOG01561, *tuf*; COG0442, *ProS*) were dropped at this stage due to the high number of transfer events, resulting in 57 single-copy orthologues for further tree inference.

## Species-tree inference

These 57 orthologous sequences were concatenated and ML trees were inferred after three independent runs with IQ-TREE 2 (ref. [86]) using the same model fitting and bootstrap settings as described above. The tree with the highest log-likelihood of the three runs was chosen as the ML species tree (topology 1). To test the effect of removing the CPR bacteria, we removed all CPR bacteria from the alignment before inferring a species tree (same parameters as above). We also performed approximately unbiased[44] tree topology tests (with IQ-TREE 2 (ref. [86]), using LG + C20 + F + G) when testing the significance of constraining the species-tree topology (ML tree; Supplementary Fig. 1) to have a DPANN clade as sister to all other Archaea (same parameters as above but with a minimally constrained topology with monophyletic Archaea and DPANN sister to other Archaea present in a polytomy (Supplementary Fig. 2)) and testing a constraint of CPR to be sister to Chloroflexi (Supplementary Fig. 3), and a combination of both the DPANN and CPR constraints (topology 2); these were tested against the ML topology, both using the normal 20 amino acid alignments and also with Susko–Roger recoding[88].

## Gene families

For the 700 representative species[15], gene family clustering was performed using EGGNOGMAPPER v.2 (ref. [89]), with the following parameters: using the DIAMOND[90] search, a query cover of 50% and an e-value threshold of 0.0000001. Gene families were collated using their KEGG[47] identifier, resulting in 9,365 gene families. These gene families were then aligned using MAFFT[84] v.7.5 with default settings and trimmed using BMGE[85] (with the same settings as above). Five independent sets of ML trees were then inferred using IQ-TREE 2 (ref. [86]), using LG + F + G, with 1,000 ultrafast bootstrap replicates. We also performed a COG-based clustering analysis in which COGs were assigned based on the modal COG identifier annotated for each KEGG gene family based on the results from EGGNOGMAPPER v.2 (ref. [89]). These gene families were aligned, trimmed and one set of gene trees (with 1,000 ultrafast bootstrap replicates) was inferred using the same parameters as described above for the KEGG gene families.

## Reconciliations

The five sets of bootstrap distributions were converted into ALE files, using ALEobserve, and reconciled against topology 1 and topology 2 using ALEml_undated[91] with the fraction missing for each genome included (where available). Gene family root origination rates were optimized for each COG functional category as previously described[35] and families were categorized into four different groups based on the probability of being present in the LUCA node in the tree. The most-stringent category was that with sampling above 1% in both domains and a PP ≥ 0.75, another category was with PP ≥ 0.75 with no sampling requirement, another with PP ≥ 0.5 with the sampling requirement; the least stringent was PP ≥ 0.5 with no sampling requirement. We used the median probability at the root from across the five runs to avoid potential biases from failed runs in the mean and to account for variation across bootstrap distributions (see Supplementary Fig. 4 for distributions of the inferred ratio of duplications, transfers and losses for all gene families across all tips in the species tree; see Supplementary Data 5 for the inferred duplications, transfers and losses ratios for LUCA, the last bacterial common ancestor and the last archaeal common ancestor).

## Metabolic pathway analysis

Metabolic pathways for gene families mapped to the LUCA node were inferred using the KEGG[47] website GUI and metabolic completeness for individual modules was estimated with Anvi'o[92] (anvi-estimate-metabolism), with pathwise completeness.

## Additional testing

We tested for the effects of model complexity on reconciliation by using posterior mean site frequency LG + C20 + F + G across three independent runs in comparison with 3 LG + F + G independent runs. We also performed a 10% subsampling of the species trees and gene family alignments across two independent runs for two different subsamples, one with and one without the presence of Asgard archaea. We also tested the likelihood of the gene families under a bacterial root (between Terrabacteria and Gracilicutes) using reconciliations of the gene families under a species-tree topology rooted as such.

## Fossil calibrations

On the basis of well-established geological events and the fossil record, we modelled 13 uniform densities to constrain the maximum and minimum ages of various nodes in our phylogeny. We constrained the bounds of the uniform densities to be either hard (no tail probability is allowed after the age constraint) or soft (a 2.5% tail probability is allowed after the age constraint) depending on the interpretation of the fossil record (Supplementary Information). Nodes that refer to the same duplication event are identified by MCMCtree as cross-braced (that is, one is chosen as the 'driver' node, the rest are 'mirrored' nodes). In other words, the sampling during the Markov chain Monte Carlo (MCMC) for cross-braced nodes is not independent: the same posterior time density is inferred for matching mirror–driver nodes (see 'Additional methods' for details on our cross-bracing approach).

## Timetree inference analyses

Timetree inference with the program MCMCtree (PAML v.4.10.7 (ref. [93])) proceeded under both the GBM and ILN relaxed-clock models. We specified a vague rate prior with the shape parameter equal to 2 and the scale parameter equal to 2.5: $\Gamma(2, 2.5)$. This gamma distribution is meant to account for the uncertainty on our estimate for the mean evolutionary rate, ~0.81 substitutions per site per time unit, which we calculated by dividing the tree height of our best-scoring ML tree (Supplementary Information) into the estimated mean root age of our phylogeny (that is, 4.520 Ga, time unit = $10^9$ years; see 'Fossil calibrations' in Supplementary Information for justifications on used calibrations). Given that we are estimating very deep divergences, the molecular clock may be seriously violated. Therefore, we applied a very diffuse gamma prior on the rate variation parameter ($\sigma^2$), $\Gamma(1, 10)$, so that it is centred around $\sigma^2 = 0.1$. To incorporate our uncertainty regarding the tree shape, we specified a uniform kernel density for the birth–death sampling process by setting the birth and death processes to 1, $\lambda$ (per-lineage birth rate) = $\mu$ (per-lineage death rate) = 1, and the sampling frequency to $\rho$ (sampling fraction) = 0.1. Our main analysis consisted of inferring the timetree for the partitioned dataset under both the GBM and the ILN relaxed-clock models in which nodes that correspond to the same divergences are cross-braced (that is, hereby referred to as cross-bracing A). In addition, we ran 10 additional inference analyses to benchmark the effect that partitioning, cross-bracing and relaxed-clock models can have on species divergence time estimation: (1) GBM + concatenated alignment + cross-bracing A, (2) GBM + concatenated alignment + cross-bracing B (only nodes that correspond to the same divergences for which there are fossil constraints are cross-braced), (3) GBM + concatenated alignment + without cross-bracing, (4) GBM + partitioned alignment + cross-bracing B, (5) GBM + partitioned alignment + without cross-bracing, (6) ILN + concatenated alignment + cross-bracing A, (7) ILN + concatenated alignment + cross-bracing B, (8) ILN + concatenated alignment + without cross-bracing, (9) ILN + partitioned alignment + cross-bracing B, and (10) ILN + partitioned alignment + without cross-bracing. Lastly, we used (1) individual gene alignments, (2) a leave-one-out strategy

(rate prior changed for alignments without *ATP* and *Leu*, Γ(2, 2.2), and without *Tyr*, Γ(2, 2.3), but was Γ(2, 2.5) for the rest; see 'Additional methods'), and (3) a more complex substitution model[94] to assess their impact on timetree inference. Refer to 'Additional methods' for details on how we parsed the dataset we used for timetree inference analyses, ran PAML programs CODEML and MCMCtree to approximate the likelihood calculation[95], and carried out the MCMC diagnostics for the results obtained under each of the previously mentioned scenarios.

### Genome size and cellular features
We simulated 100 samples of 'KEGG genomes' based on the probabilities of each of the (7,467) gene families being present in LUCA using the random.rand function in numpy[96]. The mean number of KEGG gene families was 1,298.25, the 95% HPD (highest posterior density) minimum was 1,255 and the maximum was 1,340. To infer the relationship between the number of KEGG KO gene families encoded by a genome, the number of proteins and the genome size, we used LOESS (locally estimated scatter-plot smoothing) regression to estimate the relationship between the number of KOs and (1) the number of protein-coding genes and (2) the genome size for the 700 prokaryotic genomes used in the LUCA reconstruction. To ensure that our inference of genome size is robust to uncertainty in the number of paralogues that can be expected to have been present in LUCA, we used the presence of probability for each of these KEGG KO gene families rather than the estimated copy number. We used the predict function to estimate the protein-coding genes and genome size of LUCA using these models and the simulated gene contents encoded with 95% confidence intervals.

### Additional methods
**Cross-bracing approach implemented in MCMCtree.** The PAML program MCMCtree was implemented to allow for the analysis of duplicated genes or proteins so that some nodes in the tree corresponding to the same speciation events in different paralogues share the same age. We used the tree topology depicted in Supplementary Fig. 5 to explain how users can label driver or mirror nodes (more on these terms below) so that the program identifies them as sharing the same speciation events. The tree topology shown in Supplementary Fig. 5 can be written in Newick format as:

```
(((A1,A2),A3),((B1,B2),B3));
```

In this example, A and B are paralogues and the corresponding tips labelled as A1–A3 and B1–B3 represent different species. Node *r* represents a duplication event, whereas other nodes are speciation events. If we want to constrain the same speciation events to have the same age (that is, Supplementary Fig. 5, see labels *a* and *b* (that is, A1–A2 ancestor and B1–B2 ancestor, respectively) and labels *v* and *b* (that is, A1–A2–A3 ancestor and B1–B2–B3 ancestor, respectively), we use node labels in the format #1, #2, and so on to identify such nodes:

```
(((A1, A2) #1, A3) #2, ((B1, B2) [#1 B{0.2, 0.4}],
B3) #2) 'B(0.9,1.1)';
```

Node *a* and node *b* are assigned the same label (#1) and so they share the same age (*t*): $t_a = t_b$. Similarly, node *u* and node *v* have the same age: $t_u = t_v$. The former nodes are further constrained by a soft-bound calibration based on the fossil record or geological evidence: $0.2 < t_a = t_b < 0.4$. The latter, however, does not have fossil constraints and thus the only restriction imposed is that both $t_u$ and $t_v$ are equal. Finally, there is another soft-bound calibration on the root age: $0.9 < t_r < 1.1$.

Among the nodes on the tree with the same label (for example, those nodes labelled with #1 and those with #2 in our example), one is chosen as the driver node, whereas the others are mirror nodes. If calibration information is provided on one of the shared nodes

(for example, nodes *a* and *b* in Supplementary Fig. 5), the same information therefore applies to all shared nodes. If calibration information is provided on multiple shared nodes, that information has to be the same (for example, you could not constrain node *a* with a different calibration used to constrain node *b* in Supplementary Fig. 5). The time prior (or the prior on all node ages on the tree) is constructed by using a density at the root of the tree, which is specified by the user (for example, 'B(0.9,1.1)' in our example, which has a minimum of 0.9 and a maximum of 1.1). The ages of all non-calibrated nodes are given by the uniform density. This time prior is similar to that used by ref. 29. The parameters in the birth–death sampling process (λ, μ, ρ; specified using the option BDparas in the control file that executes MCMCtree) are ignored. It is noteworthy that more than two nodes can have the same label but one node cannot have two or more labels. In addition, the prior on rates does not distinguish between speciation and duplication events. The implemented cross-bracing approach can only be enabled if option duplication = 1 is included in the control file. By default, this option is set to 0 and users are not required to include it in the control file (that is, the default option is duplication = 0).

**Timetree inference.** *Data parsing.* Eight paralogues were initially selected based on previous work showing a likely duplication event before LUCA: the amino- and carboxy-terminal regions from carbamoyl phosphate synthetase, aspartate and ornithine transcarbamoylases, histidine biosynthesis genes *A* and *F*, catalytic and non-catalytic subunits from ATP synthase (*ATP*), elongation factor Tu and G (*EF*), signal recognition protein and signal recognition particle receptor (*SRP*), tyrosyl-tRNA and tryptophanyl-tRNA synthetases (*Tyr*), and leucyl- and valyl-tRNA synthetases (*Leu*)[27]. Gene families were identified using BLASTp[97]. Sequences were downloaded from NCBI[98], aligned with MUSCLE[99] and trimmed with TrimAl[100] (-strict). Individual gene trees were inferred under the LG + C20 + F + G substitution model implemented in IQ-TREE 2 (ref. 86). These trees were manually inspected and curated to remove non-homologous sequences, horizontal gene transfers, exceptionally short or long sequences and extremely long branches. Recent paralogues or taxa of inconsistent and/or uncertain placement inferred with RogueNaRok[101] were also removed. Independent verification of an archaeal or bacterial deep split was achieved using minimal ancestor deviation[102]. This filtering process resulted in the five pairs of paralogous gene families[27] (*ATP*, *EF*, *SRP*, *Tyr* and *Leu*) that we used to estimate the origination time of LUCA. The alignment used for timetree inference consisted of 246 species, with the majority of taxa having at least two copies (for some eukaryotes, they may be represented by plastid, mitochondrial and nuclear sequences).

To assess the impact that partitioning can have on divergence time estimates, we ran our inference analyses with both a concatenated and a partitioned alignment (that is, gene partitioning scheme). We used PAML v.4.10.7 (programs CODEML and MCMCtree) for all divergence time estimation analyses. Given that a fixed tree topology is required for timetree inference with MCMCtree, we inferred the best-scoring ML tree with IQ-TREE 2 under the LG + C20 + F + G4 (ref. 103) model following our previous phylogenetic analyses. We then modified the resulting inferred tree topology following consensus views of species-level relationships[34,35,104], which we calibrated with the available fossil calibrations (see below). In addition, we ran three sensitivity tests: timetree inference (1) with each gene alignment separately, (2) under a leave-one-out strategy in which each gene alignment was iteratively removed from the concatenated dataset (for example, remove gene *ATP* but keep genes *EF*, *Leu*, *SRP* and *Tyr* concatenated in a unique alignment block; apply the same procedure for each gene family), and (3) using the vector of branch lengths, the gradient vector and the Hessian matrix estimated under a complex substitution model (bsinBV method described in ref. 94) with the concatenated dataset used for our core analyses. Four of the gene alignments generated for the leave-one-out strategy had gap-only sequences, these were removed

when re-inferring the branch lengths under the LG + C20 + F + G4 model (that is, without *ATP*, 241 species; without *EF*, 236 species; without *Leu*, 243 species; without *Tyr*, 244 species). We used these trees to set the rate prior used for timetree inference for those alignments not including *ATP*, *EF*, *Leu* or *Tyr*, respectively. The $\beta$ value (scale parameter) for the rate prior used when analysing alignments without *ATP*, *Leu* and *Tyr* changed minimally but we updated the corresponding rate priors accordingly (see above). When not including *SRP*, the alignment did not have any sequences removed (that is, 246 species). All alignments were analysed with the same rate prior, $\Gamma(2, 2.5)$, except for the three previously mentioned alignments.

*Approximating the likelihood calculation during timetree inference using PAML programs.* Before timetree inference, we ran the CODEML program to infer the branch lengths of the fixed tree topology, the gradient (first derivative of the likelihood function) and the Hessian matrix (second derivative of the likelihood function); the vectors and matrix are required to approximate the likelihood function in the dating program MCMCtree[95], an approach that substantially reduces computational time[105]. Given that CODEML does not implement the CAT (Bayesian mixture model for across-site heterogeneity) model, we ran our analyses under the closest available substitution model: LG + F + G4 (model = 3). We calculated the aforementioned vectors and matrix for each of the five gene alignments (that is, required for the partitioned alignment), for the concatenated alignment and for the concatenated alignments used for the leave-one-out strategy; the resulting values are written out in an output file called rst2. We appended the rst2 files generated for each of the five individual alignments in the same order the alignment blocks appear in the partitioned alignment file (for example, the first alignment block corresponds to the *ATP* gene alignment, and thus the first rst2 block will be the one generated when analysing the *ATP* gene alignment with CODEML). We named this file in_5parts.BV. There is only one rst2 output file for the concatenated alignments, which we renamed in.BV (main concatenated alignment and concatenated alignments under leave-one-out strategy). When analysing each gene alignment separately, we renamed the rst2 files generated for each gene alignment as in.BV.

*MCMC diagnostics.* All the chains that we ran with MCMCtree for each type of analysis underwent a protocol of MCMC diagnostics consisting of the following steps: (1) flagging and removal of problematic chains; (2) generating convergence plots before and after chain filtering; (3) using the samples collected by those chains that passed the filters (that is, assumed to have converged to the same target distribution) to summarize the results; (4) assessing chain efficiency and convergence by calculating statistics such as R-hat, tail-ESS and bulk-ESS (in-house wrapper function calling Rstan functions, Rstan v.2.21.7; https://mc-stan.org/rstan/); and (5) generating the timetrees for each type of analysis with confidence intervals and high-posterior densities to show the uncertainty surrounding the estimated divergence times. Tail-ESS is a diagnostic tool that we used to assess the sampling efficiency in the tails of the posterior distributions of all estimated divergence times, which corresponds to the minimum of the effective sample sizes for quantiles 2.5% and 97.5%. To assess the sampling efficiency in the bulk of the posterior distributions of all estimated divergence, we used bulk-ESS, which uses rank-normalized draws. Note that if tail-ESS and bulk-ESS values are larger than 100, the chains are assumed to have been efficient and reliable parameter estimates (that is, divergence times in our case). R-hat is a convergence diagnostic measure that we used to compare between- and within-chain divergence time estimates to assess chain mixing. If R-hat values are larger than 1.05, between- and within-chain estimates do not agree and thus mixing has been poor. Lastly, we assessed the impact that truncation may have on the estimated divergence times by running MCMCtree when sampling from the prior (that is, the same settings specified above

but without using sequence data, which set the prior distribution to be the target distribution during the MCMC). To summarize the samples collected during this analysis, we carried out the same MCMC diagnostics procedure previously mentioned. Supplementary Fig. 6 shows our calibration densities (commonly referred to as user-specified priors, see justifications for used calibrations above) versus the marginal densities (also known as effective priors) that MCMCtree infers when building the joint prior (that is, a prior built without sequence data that considers age constraints specified by the user, the birth–death with sampling process to infer the time densities for the uncalibrated nodes, the rate priors, and so on). We provide all our results for these quality-control checks in our GitHub repository (https://github.com/sabifo4/LUCA-divtimes) and in Extended Data Fig. 1, Supplementary Figs. 7–10 and Supplementary Data 6. Data, figures and tables used and/or generated following a step-by-step tutorial are detailed in the GitHub repository for each inference analysis.

*Additional sensitivity analyses.* We compared the divergence times we estimated with the concatenated dataset under the calibration strategy cross-bracing A with those inferred (1) for each gene, (2) for gene alignments analysed under a leave-one-out strategy, and (3) for the main concatenated dataset but when using the vector of branch lengths, the gradient vector and the Hessian matrix estimated under a more complex substitution model[94]. The results are summarized in Extended Data Fig. 2 and Supplementary Data 7 and 8. The same pattern regarding the calibration densities and marginal densities when the tree topology was pruned (that is, see above for details on the leave-one-out strategy) was observed, and thus no additional figures have been generated. As for our main analyses, the results for these additional sensitivity analyses can be found on our GitHub repository (https://github.com/sabifo4/LUCA-divtimes).

### Reporting summary
Further information on research design is available in the Nature Portfolio Reporting Summary linked to this article.

## Data availability
All data required to interpret, verify and extend the research in this article can be found at our figshare repository at https://doi.org/10.6084/m9.figshare.24428659 (ref. 106) for the reconciliation and phylogenomic analyses and GitHub at https://github.com/sabifo4/LUCA-divtimes (ref. 107) for the molecular clock analyses. Additional data are available at the University of Bristol data repository, data.bris, at https://doi.org/10.5523/bris.405xnm7ei36d2cj65nrirg3ip (ref. 108).

## Code availability
All code relating to the dating analysis can be found on GitHub at https://github.com/sabifo4/LUCA-divtimes (ref. 107), and other custom scripts can be found in our figshare repository at https://doi.org/10.6084/m9.figshare.24428659 (ref. 106).

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

## Acknowledgements

Our research is funded by the John Templeton Foundation (62220 to P.C.J.D., N.L., T.M.L., D.P., G.A.S., T.A.W. and Z.Y.; the opinions expressed in this publication are those of the authors and do not necessarily reflect the views of the John Templeton Foundation), Biotechnology and Biological Sciences Research Council (BB/T012773/1 to P.C.J.D. and Z.Y.; BB/T012951/1 to Z.Y.), by the European Research Council under the European Union's Horizon 2020 research and innovation programme (947317 ASymbEL to A.S.; 714774, GENECLOCKS to G.J.S.), Leverhulme Trust (RF-2022-167 to P.C.J.D.), Gordon and Betty Moore Foundation (GBMF9741 to T.A.W., D.P., P.C.J.D., A.S. and G.J.S.; GBMF9346 to A.S.), Royal Society (University Research Fellowship (URF) to T.A.W.), the Simons Foundation (735929LPI to A.S.) and the University of Bristol (University Research Fellowship (URF) to D.P.).

## Author contributions

The project was conceived and designed by P.C.J.D., T.M.L., D.P., G.J.S., A.S. and T.A.W. Dating analyses were performed by H.C.B., J.W.C., S.Á.-C., P.J.C.D. and E.R.R.M. T.A.M., N.D. and E.R.R.M. performed single-copy orthologue analysis for species-tree inference. L.L.S., G.J.S., T.A.W. and E.R.R.M. performed reconciliation analysis. E.R.R.M. performed homologous gene family annotation, sequence, alignment, gene tree inference and sensitivity tests. E.R.R.M., A.S. and T.A.W. performed metabolic analysis and interpretation. T.M.L., S.D. and R.A.B. provided biogeochemical interpretation. E.R.R.M., T.M.L., A.S., T.A.W., D.P. and P.J.C.D. drafted the article to which all authors (including X.C., N.L., Z.Y. and G.A.S.) contributed.

## Competing interests

The authors declare no competing interests.

## Additional information

**Extended data** is available for this paper at https://doi.org/10.1038/s41559-024-02461-1.

**Correspondence and requests for materials** should be addressed to Edmund R. R. Moody, Davide Pisani, Tom A. Williams, Timothy M. Lenton or Philip C. J. Donoghue.

[1]Bristol Palaeobiology Group, School of Earth Sciences, University of Bristol, Bristol, UK. [2]Department of Marine Microbiology and Biogeochemistry, NIOZ, Royal Netherlands Institute for Sea Research, Den Burg, The Netherlands. [3]Milner Centre for Evolution, Department of Life Sciences, University of Bath, Bath, UK. [4]Department of Biological Physics, Eötvös University, Budapest, Hungary. [5]MTA-ELTE 'Lendulet' Evolutionary Genomics Research Group, Budapest, Hungary. [6]Institute of Evolution, HUN-REN Center for Ecological Research, Budapest, Hungary. [7]Global Systems Institute, University of Exeter, Exeter, UK. [8]Department of Earth Sciences, University College London, London, UK. [9]Department of Genetics, Evolution and Environment, University College London, London, UK. [10]Model-Based Evolutionary Genomics Unit, Okinawa Institute of Science and Technology Graduate University, Okinawa, Japan. [11]Department of Evolutionary & Population Biology, Institute for Biodiversity and Ecosystem Dynamics (IBED), University of Amsterdam, Amsterdam, The Netherlands. [12]Bristol Palaeobiology Group, School of Biological Sciences, University of Bristol, Bristol, UK. ✉e-mail: edmund.moody@bristol.ac.uk; davide.pisani@bristol.ac.uk; tom.a.williams@bristol.ac.uk; T.M.Lenton@exeter.ac.uk; phil.donoghue@bristol.ac.uk

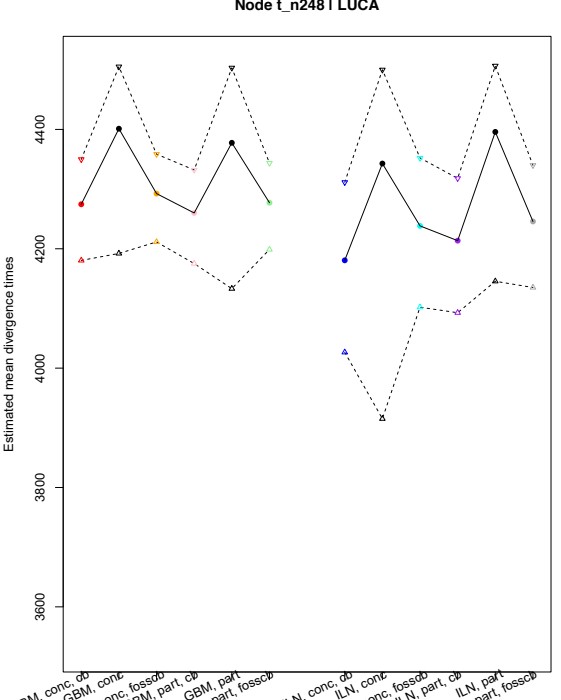

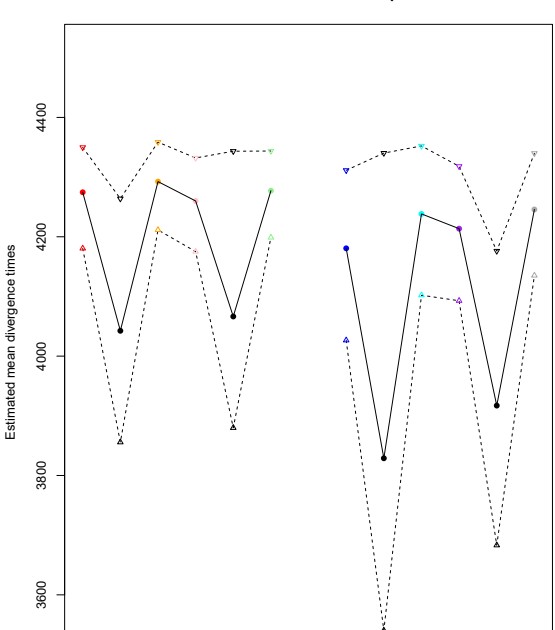

**Extended Data Fig. 1 | Comparison of the mean divergence times and confidence intervals estimated for the two duplicates of LUCA under each timetree inference analysis.** Black dots refer to estimated mean divergence times for analyses without cross-bracing, stars are used to identify those under cross-bracing and triangles for estimated upper and lower confidence intervals. Straight lines are used to link mean divergence time estimates across the various inference analyses we carried out, while dashed lines are used to link the estimated confidence intervals. The node label for the driver node is "248", while it is "368" for the mirror node, as shown in the title of each graph. Coloured stars and triangles are used to identify which LUCA time estimates were inferred under the same cross-braced analysis for the driver-mirror nodes (that is, equal time and CI estimates). Black dots and triangles are used to identify those inferred when cross-bracing was not enabled (that is, different time and CI estimates). -Abbreviations. "GBM": Geometric Brownian motion relaxed-clock model; "ILN": Independent-rate log-normal relaxed-clock model; "conc, cb" dots/triangles: results under cross-bracing A when the concatenated dataset was analysed under GBM (red) and ILN (blue); "conc, fosscb": results under cross-bracing B when the concatenated dataset was analysed under GBM (orange) and ILN (cyan); "part, cb" dots/triangles: results under cross-bracing A when the partitioned dataset was analysed under GBM (pink) and ILN (purple); "part, fosscb": results under cross-bracing B when the concatenated dataset was analysed under GBM (light green) and ILN (grey); black dots and triangles: results when cross-bracing was not enabled for both concatenated and partitioned datasets.

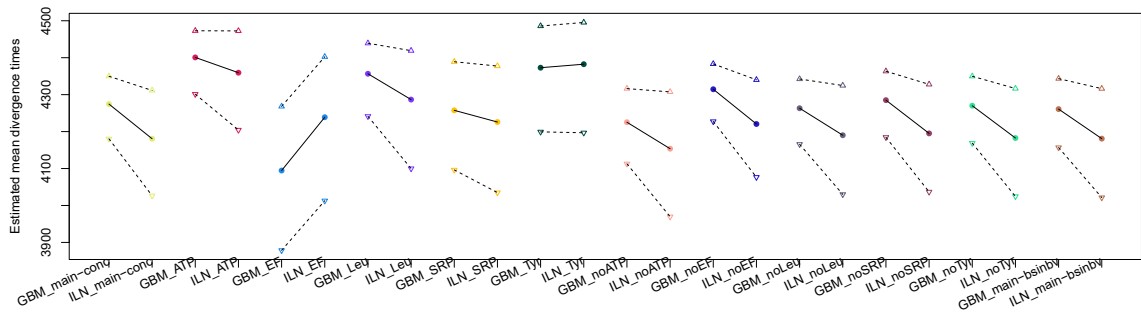

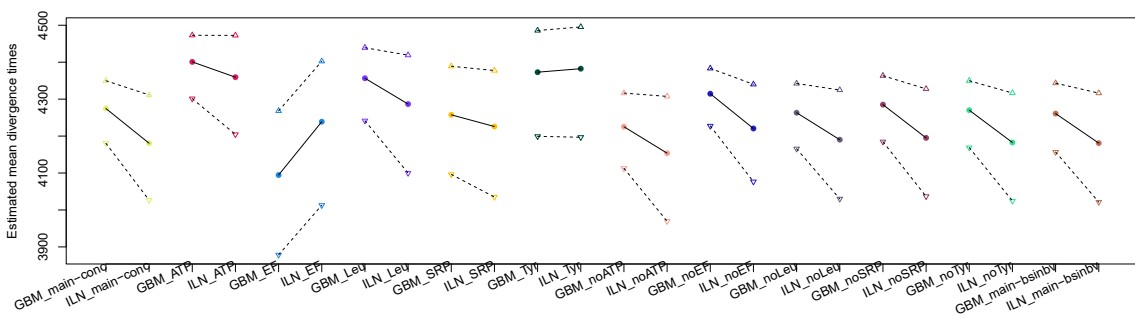

**Extended Data Fig. 2 | Comparison of the posterior time estimates and confidence intervals for the two duplicates of LUCA inferred under the main calibration strategy cross-bracing A with the concatenated dataset and with the datasets for the three additional sensitivity analyses.** Dots refer to estimated mean divergence times and triangles to estimated 2.5% and 97.5% quantiles. Straight lines are used to link the mean divergence times estimated in the same analysis under the two different relaxed-clock models (GBM and ILN). Labels in the x axis are informative about the clock model under which the analysis ran and the type of analysis we carried (see abbreviations below). Coloured dots are used to identify which time estimates were inferred when using the same dataset and strategy under GBM and ILN, while triangles refer to

the corresponding upper and lower quantiles for the 95% confidence interval. -Abbreviations. "GBM": Geometric Brownian motion relaxed-clock model; "ILN": Independent-rate log-normal relaxed-clock model; "main-conc": results obtained with the concatenated dataset analysed in our main analyses under cross-bracing A; "ATP/EF/Leu/SRP/Tyr": results obtained when using each gene alignment separately; "noATP/noEF/noLeu/noSRP/noTyr": results obtained when using concatenated alignments without the gene alignment mentioned in the label as per the "leave-one-out" strategy; "main-bsinbv": results obtained with the concatenated dataset analysed in our main analyses when using branch lengths, Hessian, and gradient calculated under a more complex substitution model to infer divergence times.

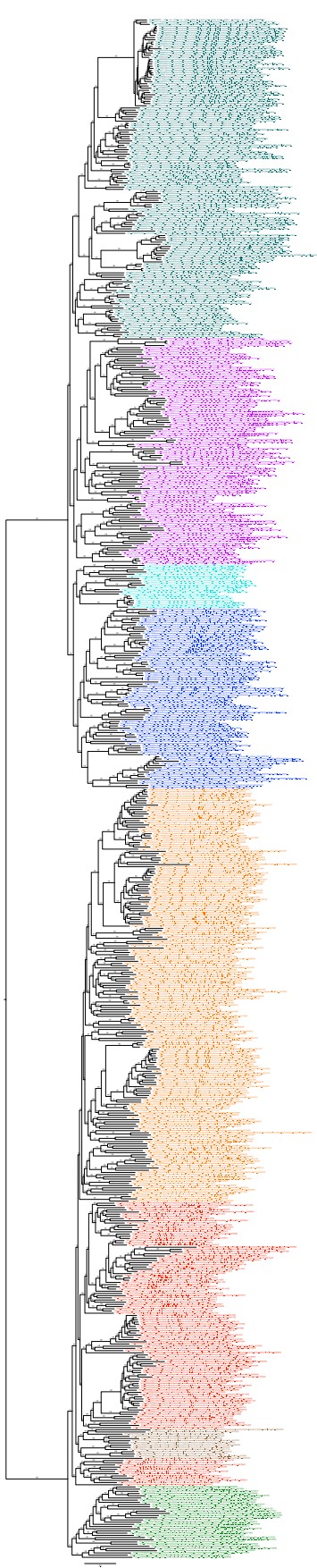

**Extended Data Fig. 3 | Maximum Likelihood species tree.** The Maximum Likelihood tree inferred across three independent runs, under the best fitting model (according to BIC: LG + F + G + C60) from a concatenation of 57 orthologous proteins, support values are from 10,000 ultrafast bootstraps.

Referred to as topology I in the main text. Tips coloured according to taxonomy: Euryarchaeota (teal), DPANN (purple), Asgardarchaeota (cyan), TACK (blue), Gracilicutes (orange), Terrabacteria (red), DST (brown), CPR (green).

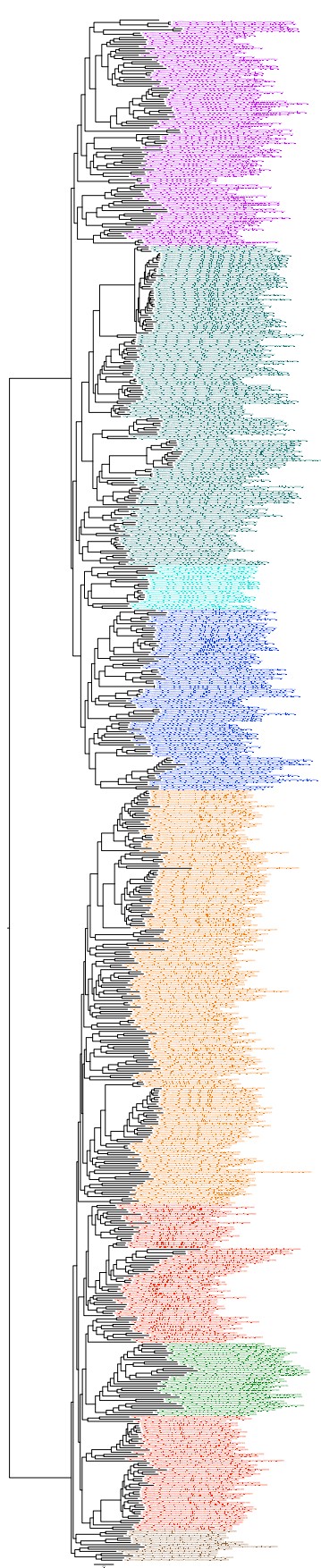

**Extended Data Fig. 4 | Maximum Likelihood tree for focal reconciliation analysis.** Maximum Likelihood tree (topology II in the main text), where DPANN is constrained to be sister to all other Archaea, and CPR is sister to Chloroflexi. Tips coloured according to taxonomy: Euryarchaeota (teal), DPANN (purple), Asgardarchaeota (cyan), TACK (blue), Gracilicutes (orange), Terrabacteria (red), DST (brown), CPR (green). AU topology test, P = 0.517, this is a one-sided statistical test.

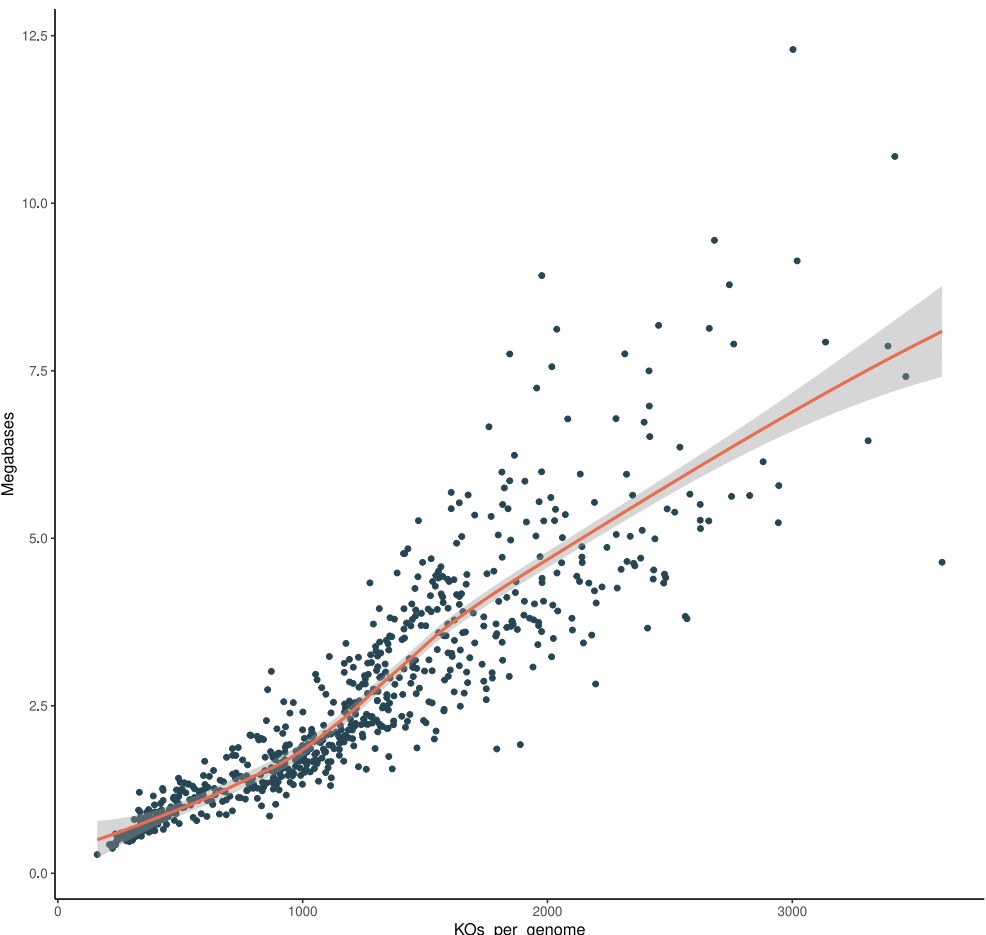

**Extended Data Fig. 5 | The relationship between the number of KO gene families encoded on a genome and its size.** LOESS regression of the number of KOs per sampled genome against the genome size in megabases. We used the inferred relationship for modern prokaryotes to estimate LUCA's genome size based on reconstructed KO gene family content, as described in the main text. Shaded area represents the 95% confidence interval.

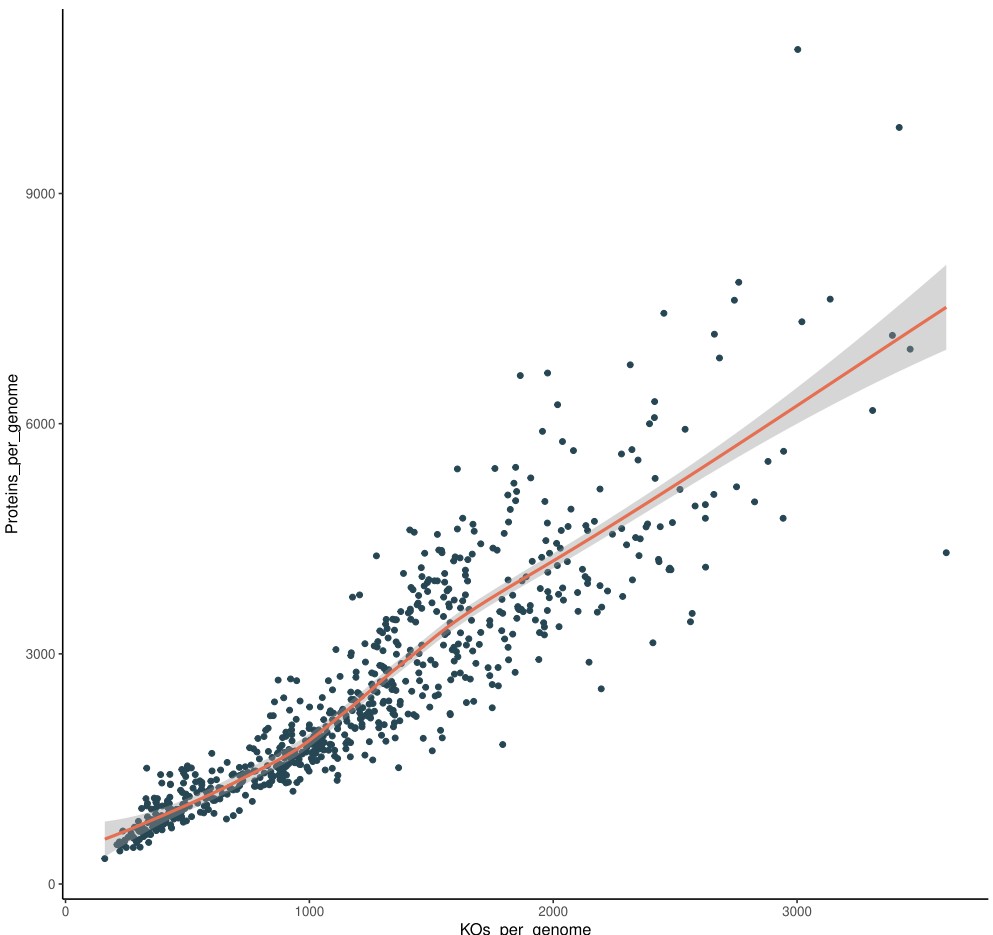

**Extended Data Fig. 6 | The relationship between the number of KO gene families encoded on a genome and the total number of protein-coding genes.** LOESS regression of the number of KOs per sampled genome against the number of proteins encoded for per sampled genome. We used the inferred relationship for modern prokaryotes to estimate the total number of protein-coding genes encoded by LUCA based on reconstructed KO gene family content, as described in the main text. Shaded area represents the 95% confidence interval.

# Reporting Summary

## Statistics

For all statistical analyses, confirm that the following items are present in the figure legend, table legend, main text, or Methods section.

| n/a | Confirmed | |
|---|---|---|
| ☒ | ☐ | The exact sample size (*n*) for each experimental group/condition, given as a discrete number and unit of measurement |
| ☒ | ☐ | A statement on whether measurements were taken from distinct samples or whether the same sample was measured repeatedly |
| ☐ | ☒ | The statistical test(s) used AND whether they are one- or two-sided <br> *Only common tests should be described solely by name; describe more complex techniques in the Methods section.* |
| ☒ | ☐ | A description of all covariates tested |
| ☒ | ☐ | A description of any assumptions or corrections, such as tests of normality and adjustment for multiple comparisons |
| ☐ | ☒ | A full description of the statistical parameters including central tendency (e.g. means) or other basic estimates (e.g. regression coefficient) AND variation (e.g. standard deviation) or associated estimates of uncertainty (e.g. confidence intervals) |
| ☐ | ☒ | For null hypothesis testing, the test statistic (e.g. *F*, *t*, *r*) with confidence intervals, effect sizes, degrees of freedom and *P* value noted <br> *Give P values as exact values whenever suitable.* |
| ☐ | ☒ | For Bayesian analysis, information on the choice of priors and Markov chain Monte Carlo settings |
| ☒ | ☐ | For hierarchical and complex designs, identification of the appropriate level for tests and full reporting of outcomes |
| ☒ | ☐ | Estimates of effect sizes (e.g. Cohen's *d*, Pearson's *r*), indicating how they were calculated |

*Our web collection on statistics for biologists contains articles on many of the points above.*

## Software and code

Policy information about availability of computer code

| Data collection | HMMER v3.3.2 <br> EGGNOGMAPPER V2 |
|---|---|
| Data analysis | BMGE 1.12 <br> MAFFT 7.407 <br> IQ-TREE 2.1.2 <br> KEGG <br> PAML 4.10.7 <br> ALEml_undated <br> R packages: rstudioapi, ape, phytools, sn, stringr, rstan, MCMCtreeRUnpublished <br> bsinBV tool: https://github.com/evolbeginner/bs_inBV/ <br> Ruby gems used by bsinBV:parallel, colorize, find, getoptlong, time, fileutils, tmpdir, bio, bio-nwk <br> Third party software required by bsinBV: newick-utils v1.6 |

For manuscripts utilizing custom algorithms or software that are central to the research but not yet described in published literature, software must be made available to editors and reviewers. We strongly encourage code deposition in a community repository (e.g. GitHub). See the Nature Portfolio guidelines for submitting code & software for further information.

# Data

Policy information about availability of data

All manuscripts must include a data availability statement. This statement should provide the following information, where applicable:

- Accession codes, unique identifiers, or web links for publicly available datasets
- A description of any restrictions on data availability
- For clinical datasets or third party data, please ensure that the statement adheres to our policy

> All alignments (with accession numbers), reconciliations tree-files and other data and results files can be found at our figshare repository:10.6084/m9.figshare.24428659. [Private link for reviewers  https://figshare.com/s/4f4ab42f6a2e0886a161

# Research involving human participants, their data, or biological material

Policy information about studies with human participants or human data. See also policy information about sex, gender (identity/presentation), and sexual orientation and race, ethnicity and racism.

| | |
|---|---|
| Reporting on sex and gender | *Use the terms sex (biological attribute) and gender (shaped by social and cultural circumstances) carefully in order to avoid confusing both terms. Indicate if findings apply to only one sex or gender; describe whether sex and gender were considered in study design; whether sex and/or gender was determined based on self-reporting or assigned and methods used. Provide in the source data disaggregated sex and gender data, where this information has been collected, and if consent has been obtained for sharing of individual-level data; provide overall numbers in this Reporting Summary.  Please state if this information has not been collected. Report sex- and gender-based analyses where performed, justify reasons for lack of sex- and gender-based analysis.* |
| Reporting on race, ethnicity, or other socially relevant groupings | *Please specify the socially constructed or socially relevant categorization variable(s) used in your manuscript and explain why they were used. Please note that such variables should not be used as proxies for other socially constructed/relevant variables (for example, race or ethnicity should not be used as a proxy for socioeconomic status). Provide clear definitions of the relevant terms used, how they were provided (by the participants/respondents, the researchers, or third parties), and the method(s) used to classify people into the different categories (e.g. self-report, census or administrative data, social media data, etc.) Please provide details about how you controlled for confounding variables in your analyses.* |
| Population characteristics | *Describe the covariate-relevant population characteristics of the human research participants (e.g. age, genotypic information, past and current diagnosis and treatment categories). If you filled out the behavioural & social sciences study design questions and have nothing to add here, write "See above."* |
| Recruitment | *Describe how participants were recruited. Outline any potential self-selection bias or other biases that may be present and how these are likely to impact results.* |
| Ethics oversight | *Identify the organization(s) that approved the study protocol.* |

Note that full information on the approval of the study protocol must also be provided in the manuscript.

# Field-specific reporting

Please select the one below that is the best fit for your research. If you are not sure, read the appropriate sections before making your selection.

☒ Life sciences ☐ Behavioural & social sciences ☐ Ecological, evolutionary & environmental sciences

For a reference copy of the document with all sections, see nature.com/documents/nr-reporting-summary-flat.pdf

# Life sciences study design

All studies must disclose on these points even when the disclosure is negative.

| | |
|---|---|
| Sample size | Samples were taken from previously published datasets |
| Data exclusions | No data was excluded |
| Replication | Final results for LUCA PPs were taken from multiple independent replications of bootstraps, reconciled with both constrained and unconstrained species tree as detailed in the text. |
| Randomization | Subsampling for additional tests is explained in the supplementary material, this was done none randomly by using the highest genome quality per taxonomic rank |
| Blinding | Blinding not relevant as this is primarily a phylogenomic analyses of prokaryotes |

# Reporting for specific materials, systems and methods

We require information from authors about some types of materials, experimental systems and methods used in many studies. Here, indicate whether each material, system or method listed is relevant to your study. If you are not sure if a list item applies to your research, read the appropriate section before selecting a response.

## Materials & experimental systems

| n/a | Involved in the study |
|-----|----------------------|
| ☒ ☐ | Antibodies |
| ☒ ☐ | Eukaryotic cell lines |
| ☒ ☐ | Palaeontology and archaeology |
| ☒ ☐ | Animals and other organisms |
| ☒ ☐ | Clinical data |
| ☒ ☐ | Dual use research of concern |
| ☒ ☐ | Plants |

## Methods

| n/a | Involved in the study |
|-----|----------------------|
| ☒ ☐ | ChIP-seq |
| ☒ ☐ | Flow cytometry |
| ☒ ☐ | MRI-based neuroimaging |

## Plants

| | |
|---|---|
| Seed stocks | *Report on the source of all seed stocks or other plant material used. If applicable, state the seed stock centre and catalogue number. If plant specimens were collected from the field, describe the collection location, date and sampling procedures.* |
| Novel plant genotypes | *Describe the methods by which all novel plant genotypes were produced. This includes those generated by transgenic approaches, gene editing, chemical/radiation-based mutagenesis and hybridization. For transgenic lines, describe the transformation method, the number of independent lines analyzed and the generation upon which experiments were performed. For gene-edited lines, describe the editor used, the endogenous sequence targeted for editing, the targeting guide RNA sequence (if applicable) and how the editor was applied.* |
| Authentication | *Describe any authentication procedures for each seed stock used or novel genotype generated. Describe any experiments used to assess the effect of a mutation and, where applicable, how potential secondary effects (e.g. second site T-DNA insertions, mosiacism, off-target gene editing) were examined.* |

