## [Peer Review File · Nature Ecology & Evolution]

Peer Review Information

Journal: Nature Ecology & Evolution

Manuscript Title: The nature of the Last Universal Common Ancestor and its impact on the early Earth system

Corresponding author name(s): Edmund R. R. Moody, Davide Pisani, Tom A. Williams, Timothy M. Lenton, Philip C. J. Donoghue

Editorial Notes:

Reviewer Comments & Decisions:

Decision Letter, initial version:

20th February 2024

Dear Ed,

Your manuscript entitled "The nature of the Last Universal Common Ancestor and its impact on the early Earth system" has now been seen by two reviewers, whose comments are attached. The reviewers have raised a number of concerns which will need to be addressed before we can offer publication in Nature Ecology & Evolution. We will therefore need to see your responses to the criticisms raised and to some editorial concerns, along with a revised manuscript, before we can reach a final decision regarding publication.

We therefore invite you to revise your manuscript taking into account all reviewer and editor comments. Please highlight all changes in the manuscript text file in Microsoft Word format.

* If you have not done so already please begin to revise your manuscript so that it conforms to our Article format instructions at <http://www.nature.com/natecolevol/info/final-submission>. Refer also to any guidelines provided in this letter.

2[REDACTED]

Nature Ecology & Evolution is committed to improving transparency in authorship. As part of our efforts in this direction, we are now requesting that all authors identified as 'corresponding author' on published papers create and link their Open Researcher and Contributor Identifier (ORCID) with their account on the Manuscript Tracking System (MTS), prior to acceptance. ORCID helps the scientific community achieve unambiguous attribution of all scholarly contributions. You can create and link your ORCID from the home page of the MTS by clicking on 'Modify my Springer Nature account'. For more information please visit please visit www.springernature.com/orcid.

[REDACTED]

Reviewer expertise:

Reviewer #1: microbial early life, phylogenetics and metabolism reconstruction

Reviewer #2: microbial early life, phylogenetics

Reviewers' comments:

Reviewer #1 (Remarks to the Author):

In their manuscript, "The Nature of the Last Universal Common Ancestor and Its Impact on the Early Earth System", Moody and colleagues present a detailed description of the Last Universal Common Ancestor (LUCA), the time in which it lived, properties of its genome and proteome, and its environment and ecological community. The study concludes that the LUCA lived roughly 4.2 Ga, earlier than previous estimates, that the LUCA had a genome and proteome size similar to modern prokaryotes, and that it lived in one of several possible environmental niches alongside other organisms upon which it was metabolically dependent.

2This manuscript represents a major advance in our understanding of the LUCA and will garner broad interest across and beyond the early evolution field. By using tree reconciliation to predict the proteome of the LUCA, the authors are able to present a probable LUCA proteome rather than a minimal LUCA proteome and use these probabilities to estimate the genome and proteome size of the LUCA. This tree reconciliation approach, which the authors have successfully applied to the ancestors of the archaea and bacteria in previous publications, represents a level of rigor that is rare in LUCA proteome reconstructions. This is only one of several innovations presented in the article. Another is the use of universal paralog families to reinforce a time tree analysis that credibly dates the LUCA. And yet another innovation is the use of the predicted LUCA proteome to infer the metabolic community in which the LUCA lived as well as the presence of viruses that it defended itself against. Taken together, I would not be surprised if this article was received as a breakthrough in the discipline. Even so, I have some questions and comments that I hope the authors can address.

Major comments:

- My biggest concern is whether tree reconciliation is accurate at this deep of a node. My own lab has used ALE to assess LUCA ancestry and to estimate DTL events and I have found that, for protein families that likely originated in the LUCA, ALE will sometimes infer a large number of duplications and losses prior to the LUCA and also on the bacterial and archaea stems. I have always taken this as a sign that ALE did not produce a trustworthy prediction. I would not be surprised if ALE performed better in these authors' hands (as some of them wrote ALE in the first place), but I am curious if the authors see a similar pattern of multiple duplications and losses before the LUCA and in the bacterial/archaeal stems and, if so, whether they interpret such results differently than I have.

- Estimating the protein content of the LUCA by summing the PPs is mathematically correct, but it seems too sensitive to the number of protein families being considered. Looking at the supplemental information, most protein families that were subjected to the analysis have a PP of >0.065 (this is the median PP), which is a small value but would add up quickly. If the most conservative estimate of 2451 proteins is applied to the distribution of PPs, this would mean that proteins with a $PP \leq 0.01$ must be included in the LUCA proteome. These issues should be discussed alongside the estimate of the proteome size.

- The universal paralog families used for the time tree analysis are based on the Zhaxybayeva et al. (2005) study. This study found that different paralog trees (and sometimes clades within the same paralog tree) suggest different species tree topologies. How did this study take different paralog tree topologies into account when dating these ancestors on a single species tree (as indicated in Figure 1)?

Minor Comments:

- Line 103: The pre-LUCA duplication of the ATP synthase was between the catalytic subunit and non-catalytic subunit of the F-type ATPase, not between the F- and V-type ATPases. I assume this was just a mislabeling or the authors would have noticed that their tree was not a true universal paralog.

- Lines 158-160: This method is not well described in the Methods section or the supplement, but I

3assume that the use of "modern genomes as training data" was to create a model for converting the size of the proteome to a genome size. Is this correct? If so, did the set of genomes used to train the model exclude eukaryotes (considering the relationship between proteome size and genome size is very different in eukaryotes)?

- Lines 291-308: Based on metabolic analysis of the probable LUCA proteome, the LUCA is inferred to have either lived in a deep sea hydrothermal vent niche or a shallow ocean niche. The authors then make the leap that the LUCA "could have occupied both niches i.e. a shallow hydrothermal vent" (lines 307 and 308). But this is really be a third niche which is just as plausible an environment as either a deep sea hydrothermal vent niche or a shallow ocean niche. Actually, it is less likely of a LUCA environment because there would be fewer niches that satisfy both criteria than niches that satisfy one of them (i.e., a conjunction fallacy).

- Lines 354-364: This paragraph seems to mostly repeat the concluding paragraph of the previous section (lines 257/354, 260/357, 265/359). It reads like a very brief conclusion of the entire paper rather than a concluding paragraph to the "LUCA's Environment, Ecosystem and Earth system context" section. If this was the authors' intent, I would recommend adding a Conclusions section. Otherwise, I would recommend removing this paragraph.

- Supplement: There should be an explanation of all of the supplemental data files along with the meaning of the table headers either in the main text of the manuscript or as an additional supplemental file.

Reviewer #2 (Remarks to the Author):

Moody et al., have investigated the age and gene repertoire of LUCA, and based on their findings have interpreted a plausible ecological context for LUCA. These are certainly issues of great importance to understanding the most fundamental aspects of life's origin and cell evolution. The study uses a set of recently developed methods, and this is much welcomed (e.g., cross-bracing for node dating). The use of gene-tree—species-tree reconciliation approach is a clear advantage over previous efforts for estimating ancestral gene contents. The interpretation of the results and the conclusions offered are reasonable.

In L69-71, the authors state "Relaxed Bayesian node-calibrated molecular clock approaches provide a means of integrating the sparse fossil record of early life with the information provided by extant genomes." However, the genome information the authors were able to use in their analysis is a rather small fraction which raises the concern of how much phylogenetic information has been retained in those few genes to accurately infer genetic distances and divergence times for the most ancient events in life's history. Why did the authors limit their analysis to only five paralog pairs? This selection is not explained in the text.

Even though the authors have used a new and interesting method, i.e., cross-bracing on pre-LUCA paralogs, this also comes with some drawbacks that have to be explained and justified. For example, CODEML does not incorporate site-heterogenous models of protein sequence evolution. This can

4further affect the inference of more realistic genetic distances and divergence times.

In light of recent criticisms on the limitations of molecular dating

(<https://doi.org/10.1093/sysbio/syad057> and <https://doi.org/10.1038/d41586-023-03487-4>), I think the authors have to further explain/justify and elaborate on why their estimated ages for LUCA (and other nodes) are not artefacts.

Minor comments:

L82-85: Sentence does not read well. Revise.

L102-106: Why is the dataset limited to these five paralog pairs? Are there not more paralogs that predate LUCA?

L158-160: Reads awkwardly.

How do the inferred probabilities change with taxon sample size? It would be interesting to see how robust the methods are to this.

Check the spelling of Wood-Ljungdahl throughout the text.

L275: Please provide some examples of these alternative electron donors.

L239: Explain what the function of the ffh gene is.

L361: Change to origin of life.

Fig. 1: Depict/annotate fossil calibrations used.

*****END*****

Author Rebuttal to Initial comments

Response to reviewers

Reviewer #1 (Remarks to the Author):

In their manuscript, "The Nature of the Last Universal Common Ancestor and Its Impact on the Early Earth System", Moody and colleagues present a detailed description of the Last Universal Common Ancestor (LUCA), the time in which it lived, properties of its genome and proteome, and its environment and ecological community. The study concludes that the LUCA lived roughly 4.2 Ga, earlier than previous estimates, that the LUCA had a genome and proteome size similar to modern prokaryotes, and that it lived in one of several possible environmental niches alongside other organisms upon which it was metabolically dependent.

This manuscript represents a major advance in our understanding of the LUCA and will garner broad interest across and beyond the early evolution field. By using tree reconciliation to predict the proteome of the LUCA, the authors are able to present a probable LUCA proteome rather than a minimal LUCA proteome and use these probabilities to estimate the genome and proteome size of the LUCA. This tree reconciliation approach, which the authors have successfully applied to the ancestors of the archaea and bacteria in previous publications, represents a level of rigor that is rare in LUCA proteome reconstructions. This is only one of several innovations presented in the article. Another is the use of universal paralog families to reinforce a time tree analysis that credibly dates the LUCA. And yet another innovation is the use of the predicted LUCA proteome to infer the metabolic community in which the LUCA lived as well as the presence of viruses that it defended itself against. Taken together, I would not be surprised if this article was received as a breakthrough in the discipline. Even so, I have some questions and comments that I hope the authors can address.

Major comments:

- My biggest concern is whether tree reconciliation is accurate at this deep of a node. My own lab has used ALE to assess LUCA ancestry and to estimate DTL events and I have found that, for protein families that likely originated in the LUCA, ALE will sometimes infer a large number of duplications and losses prior to the LUCA and also on the bacterial and archaea stems. I have always taken this as a sign that ALE did not produce a trustworthy prediction. I would not be surprised if ALE performed better in these authors' hands (as some of them wrote ALE in the first place), but I am curious if the authors see a similar pattern of multiple duplications and losses before the LUCA and in the bacterial/archaeal stems and, if so, whether they interpret such results differently than I have.

We thank the reviewer for this comment. To investigate whether ALE inferred an excessive number of duplications or losses in LUCA or on the archaeal or bacterial stems, we counted the inferred number of duplications, transfers and losses for each node and tip in our species

tree, for each gene family (using the median value over the five independent bootstrap distributions as the estimate in each case). In short, the inferred numbers of events on the branches leading to LUCA, LACA and LBCA are unexceptional, falling well within the range of branchwise values seen across the prokaryotic tree of life (see table below, also provided as Supplementary Table 5). One interesting exception is the number of duplications on the archaeal stem, which is in the top 5% of branchwise numbers of events overall, but not the highest number across all nodes. Manual inspection of inferred gene trees for the top five gene families for duplications along the archeal stem (i.e. K10896 (Hef), K01873 (valyl-tRNA synthetase), K00584 (mtrH); tetrahydromethanopterin S-methyltransferase subunit H), K04795 (fibrillar-like protein), K07321 (CO dehydrogenase maturation factor, ATPase), K00555 (tRNA(guanine 26, N (2), N (2))-dimethyltransferase)) would suggest that these genes have duplicated by this point, suggesting the signal captured by ALE reflects evidence in the underlying gene trees.

We have included histograms of these distributions in an updated supplementary plot (Supplementary Figure 4), as this is an important and useful point to make, and below is a table of the estimated values for LBCA, LACA and LUCA, the mean ratios across all nodes in the tree and the 95th percentiles.

Node	Ratio of event to total copy number		
	Duplications	Transfers	Losses
LUCA	0.0352	0	0.0465
LACA	0.101	0.164	0.285
LBCA	0.0295	0.240	0.685
Mean (all nodes)	0.02720271	0.2288356	0.4031224
95th Percentile	0.07548088	0.4133611	0.8534644

- Estimating the protein content of the LUCA by summing the PPs is mathematically correct, but it seems too sensitive to the number of protein families being considered. Looking at the supplemental information, most protein families that were subjected to the analysis have a PP of >0.065 (this is the median PP), which is a small value but would add up quickly. If the most conservative estimate of 2451 proteins is applied to the distribution of PPs, this would mean that proteins with a PP \leq 0.01 must be included in the LUCA proteome. These issues should be discussed alongside the estimate of the proteome size.

Thanks for pointing this out - we agree that straightforwardly summing the PPs would face this issue. On reflection, we should have been clearer about how we calculated genome size estimates from the per-family PPs in the first version of the manuscript. We did not directly sum the PPs, but rather generated estimates of ancestral gene counts by

sampling from each gene family according to the PP; the range of sizes reflects the variation in total gene copy numbers obtained by repeating this procedure 100 times. This procedure ameliorates the reviewer's concern, in the sense that most families with, for example, $PP < 0.01$, will not be included in any particular simulated gene count. We also note that the relationship between the number of KO gene families and genome size estimated for extant taxa was performed using only the same set of gene families we analyse in the study, which further helps to correct the analysis for the number of gene families analysed.

We have now also repeated this analysis based on the COG-based gene families; the results (now also included in the supplementary) infer a slightly smaller genome size for LUCA (1.45+ Mb), with a similarly decreased number of proteins (1466+) when using COG gene families. We have now made this more explicit in the methods, and have also revised previous references to the summed total of (KEGG) gene families (1297) to refer to the mean number of KEGG gene families (1298.25) from across our simulations to make this clearer in the main text and the methods.

We now write, in the main text:

"By using modern prokaryotic genomes as training data, we used a predictive model to estimate the genome size, and the number of protein families LUCA encoded, based on the relationship between the number of KEGG gene families and total number of proteins encoded by modern prokaryote genomes (Extended Data Fig. 3 and 4). Our predictive model estimates a genome size of 2.75Mb (2.49Mb - 2.99Mb) encoding 2657 (2451 - 2855) proteins (see Methods). While we can estimate the number of genes in LUCA's genome, it is more difficult to identify the specific gene families that might have already been present in LUCA, based on the genomes of modern Archaea and Bacteria. It is likely that the modern version of the pathways would be considered incomplete based on LUCA's gene content through subsequent evolutionary changes. "

And in the methods:

"We simulated 100 samples of 'KEGG genomes' based on the probabilities of each of the (7467) gene families being present in LUCA using the 'random.rand' function in numpy97. The mean number of KEGG gene families was 1298.25, and the 95% HPD minimum: 1255, and maximum: 1340. To infer the relationship between the number of KEGG KO gene families encoded by a genome, the number of proteins, and the genome size, we used LOESS regression to estimate the relationship between number of KOs and (i) number of protein-coding genes, (ii) genome size for the 700 prokaryotic genomes used in the LUCA reconstruction. We used the predict function to estimate the protein-coding genes and genome size of LUCA using these models and the simulated gene contents encoded with 95% confidence intervals."

- The universal paralog families used for the time tree analysis are based on the Zhaxybayeva et al. (2005) study. This study found that different paralog trees (and sometimes clades within the same paralog tree) suggest different species tree topologies. How did this study take different paralog tree topologies into account when dating these ancestors on a single species tree (as indicated in Figure 1)?

We only used pre-LUCA duplicates that support a position for LUCA between Bacteria and Archaea, but as the reviewer points out, there are some minor differences in species tree topologies between paralogues (and gene families). However these differences are not unexpected given that single gene trees often lack the signal to fully resolve ancient relationships and given that the duplications in these genes are extremely old. In our original analyses, we treated these differences as random errors (expected in individual gene analyses) and dated a fixed species tree which topology was based on consensus views of species-level relationships, even if it did not completely agree with individual topologies that could have been inferred for all the paralogues in all gene trees. However, we agree with the reviewer that it is important to quantify the potential impact that including different paralogues in the alignment can have on timetree inference, specifically on our LUCA time estimates.

To address this concern, we have performed additional timetree inference analyses with (i) individual pairs of paralogues (divergence times estimated for each of the five gene alignments separately) and (ii) concatenated alignments following a "leave-one-out" strategy. For individual pairs, LUCA time estimates inferred under GBM were relatively younger than those under ILN, with one exception (for the Elongation Factor paralogue pair the reverse was true). Under the "leave-one-out" strategy (for each of the five paralogue pairs, we excluded one of these pairs from a concatenation of the remaining four) the same pattern was observed for four of the five concatenations, regardless of the clock, with one exception (estimates inferred when removing the ATP synthase pair were relatively younger).

The LUCA time estimates for our focal analyses with the concatenated alignment and cross-bracing would seem to account for the differences we have observed in our additional timetree inference analyses: both the mean divergence times and confidence intervals are an average over those obtained when analysing individual gene trees and the concatenated alignments we assembled under the "leave-one-out" strategy aforementioned. We have summarised our latest results in Extended Figure Data 6 so that they are easier to interpret.

Minor Comments:

- Line 103: The pre-LUCA duplication of the ATP synthase was between the catalytic subunit and non-catalytic subunit of the F-type ATPase, not between the F- and V-type ATPases. I assume this was just a mislabeling or the authors would have noticed that their tree was not a true universal paralog.

We thank the reviewers for pointing out this mislabelling. They are indeed the catalytic and non-catalytic subunits. We have corrected this in the text by changing 'F- and V-type', to 'catalytic and non-catalytic'.

- Lines 158-160: This method is not well described in the Methods section or the supplement, but I assume that the use of "modern genomes as training data" was to create a model for converting the size of the proteome to a genome size. Is this correct? If so, did the set of genomes used to train the model exclude eukaryotes (considering the relationship between proteome size and genome size is very different in eukaryotes)?

We have now made the methods section clearer in line with this comment and the major comment above. We excluded eukaryotes for this reason and now explicitly state this as advised (see comment above for revised methods section).

- Lines 291-308: Based on metabolic analysis of the probable LUCA proteome, the LUCA is inferred to have either lived in a deep sea hydrothermal vent niche or a shallow ocean niche. The authors then make the leap that the LUCA "could have occupied both niches i.e. a shallow hydrothermal vent" (lines 307 and 308). But this is really be a third niche which is just as plausible an environment as either a deep sea hydrothermal vent niche or a shallow ocean niche. Actually, it is less likely of a LUCA environment because there would be fewer niches that satisfy both criteria than niches that satisfy one of them (i.e., a conjunction fallacy).

This is a good point and we have revised the text accordingly, stating:

"Previous studies often favoured a deep ocean environment for LUCA as early life would have been better protected there from an episode of LHB. However, if LHB was less intense than initially proposed^{21,23}, or else just a sampling artefact²², these arguments weaken. Another possibility, may be that LUCA inhabited a shallow hydrothermal vent or a hot spring."

- Lines 354-364: This paragraph seems to mostly repeat the concluding paragraph of the previous section (lines 257/354, 260/357, 265/359). It reads like a very brief conclusion of the entire paper rather than a concluding paragraph to the "LUCA's Environment, Ecosystem and Earth system context" section. If this was the authors' intent, I would recommend adding a Conclusions section. Otherwise, I would recommend removing this paragraph.

We have now modified this to be a separate conclusions section, as advised.

- Supplement: There should be an explanation of all of the supplemental data files along with the meaning of the table headers either in the main text of the manuscript or as an additional supplemental file.

We thank the reviewer for this comment and have made the supplemental table figure captions at the end of the supplementary material clearer. We have also included an expanded explanation of the supplemental data files in the data repository readme file.

Reviewer #2 (Remarks to the Author):

Moody et al., have investigated the age and gene repertoire of LUCA, and based on their findings have interpreted a plausible ecological context for LUCA. These are certainly issues of great importance to understanding the most fundamental aspects of life's origin and cell evolution. The study uses a set of recently developed methods, and this is much welcomed (e.g., cross-bracing for node dating). The use of gene-tree–species-tree reconciliation approach is a clear advantage over previous efforts for estimating ancestral gene contents. The interpretation of the results and the conclusions offered are reasonable.

In L69-71, the authors state “Relaxed Bayesian node-calibrated molecular clock approaches provide a means of integrating the sparse fossil record of early life with the information provided by extant genomes.” However, the genome information the authors were able to use in their analysis is a rather small fraction which raises the concern of how much phylogenetic information has been retained in those few genes to accurately infer genetic distances and divergence times for the most ancient events in life's history. Why did the authors limit their analysis to only five paralog pairs? This selection is not explained in the text.

This is a good point, unfortunately there are only a limited number of pre-LUCA gene duplications that can be unambiguously rooted to separate Archaea and Bacteria, as shown in Zhaxybayeva *et al.* (2005). We initially investigated all eight genes identified by Zhaxybayeva *et al.* (2005). However, our re-analyses indicated that only five of these eight genes could be unambiguously rooted to separate Bacteria and Archaea. For three of these eight genes (the N and C terminal regions from carbamoylphosphate synthetase; aspartate and ornithine transcarbamoylases; histidine biosynthesis genes A and F), an unambiguous rooting point separating all Archaea from all Bacteria could not be identified. To avoid introducing errors in our estimate of the age of LUCA, we opted to avoid using these three gene families.

In response to this comment and comments from the reviewer above we have performed several additional analyses which indicate that the five retained gene families are sufficiently informative to generate useful insights on the age of LUCA: firstly, both individual and concatenated analyses of these genes infer timetrees broadly consistent with current evidence from species trees based on concatenation. Secondly, the credibility intervals associated with our inferred divergence times, despite being wide, are not unreasonably broad, indicating the presence of signal in the data.

Our individual pair and “leave-one-out” analyses returned results indicating that the exclusion of “ATP” from a concatenated alignment and the analysis of “EF” as a single-gene alignment result in somewhat younger time estimates regardless of the clock used. The time estimates obtained with the concatenated alignment under cross-bracing fall within the differences we see when analysing these individual gene alignments or when pairs are excluded from the analysis. Overall, these results suggest the five paralogue pairs we used for timetree inference are informative and have not introduced biases. We have amended the text to better explain our methodology.

Even though the authors have used a new and interesting method, i.e., cross-bracing on pre-LUCA paralogs, this also comes with some drawbacks that have to be explained and justified. For example, CODEML does not incorporate site-heterogeneous models of protein sequence evolution. This can further affect the inference of more realistic genetic distances and divergence times.

In light of recent criticisms on the limitations of molecular dating (<https://doi.org/10.1093/sysbio/syad057> and <https://doi.org/10.1038/d41586-023-03487-4>), I think the authors have to further explain/justify and elaborate on why their estimated ages for LUCA (and other nodes) are not artefacts.

Thanks for raising these points. With regard to the point about site-heterogeneous substitution models, in our revision we have now performed dating analyses using the best-fitting site-heterogeneous substitution model (LG+C60+F+G), using the approach recently developed by Wang and Luo. (preprint, <https://www.biorxiv.org/content/10.1101/2023.06.18.545440v1>) to calculate the branch lengths, gradient, and Hessian (used by MCMCTree to approximate the likelihood) under complex substitution models (see their GitHub repository: https://github.com/evolbeginner/bs_inBV). We adapted the publicly available scripts so that it could work in a different environment than the one used by the authors (see our comments suggesting the changes we had to make under the “Issues” section https://github.com/evolbeginner/bs_inBV/issues).

The LUCA time estimates (as well as other time estimates) we obtained are not significantly different from those we originally obtained when using the branch lengths, the gradient, and the Hessian estimated under “LG+F+G4” for timetree inference. Please note that LUCA is not the root of our tree topology. The root time estimate is the one that is mainly affected when using more simple protein substitution models as reported by the simulations ran by Wang and Luo (preprint), which is not the case in this study. Our results are summarised in our Extended Figure Data 6.

With regard to the second point - the recent, rather general critique of molecular clock methods by Budd and Mann (2023) - a full analysis of their claims would be beyond the scope of the current paper. We briefly summarise why we are not convinced by their reasoning below, and in the main text we mention this debate while discussing the difficulty of dating the root of the tree, and so the motivation for the paralogue-dating approach.

In Budd and Mann (2023), the authors base their criticism on their interpretation of the fossil record and their understanding of calibration densities and marginal densities (also known as user-specified priors and effective priors, respectively) widely used in Bayesian node-dating analyses. Nevertheless, they fail to understand what the joint prior for all times is and how marginal densities are estimated by dating programs (not only MCMCtree, although they specifically refer to such program), which misleads readers to think that (i) marginal densities do not capture the uncertainty of the fossil record they initially specified as calibration densities and (ii) the results obtained in molecular-clock dating analyses are artefacts.

Particularly, the authors (Budd and Mann, 2023) do not account for the fact that there are various uncalibrated internal nodes relative to those where the age is being constrained by calibration densities specified by users (either daughter or parental nodes). Based on such calibration densities and the birth-death with sampling process, the dating program (MCMCtree in our case) will infer the time densities that will be assigned to the uncalibrated nodes. Please note that neither of these time densities is the joint prior of all times. The time prior is a complex high-dimensional distribution that, unfortunately, cannot be specified by the users (although that would be the ideal scenario!). Instead, users specify calibration densities to constrain specific node ages in their tree topologies (based on knowledge on the fossil record, biomarkers, geological events, etc.) and the parameter values of the birth-death with sampling process (μ , λ , and ρ). To that end, the dating program will use such information to derive the time densities of the uncalibrated nodes which, together with calibration densities, the root age constraint, and the birth-death with sampling process; will generate the joint prior of all times. Once this is understood, it is easy to see why, oftentimes, calibration densities may differ from the corresponding marginal densities – note that all users should always examine such densities (i.e., run the program without data) and make sure they are sensible before running the dating programs with their datasets (advice given not only in the PAML documentation to run MCMCtree and the documentation of other dating programs, but also in many other papers such as those criticised by Budd and Mann).

Once the joint prior has been inferred, it is possible that some regions of calibration densities specified for a given node may now be “occupied” by densities assigned to uncalibrated neighbouring parental/daughter nodes. Consequently, marginal densities may end up differing from calibration densities – but this does not mean that the information from the fossil record or other knowledge used to build the calibration density has not been used! To illustrate this situation, we have drawn a simple example with a uniform calibration density with soft-bounds (red) specified by the user to constrain the age of a node (red star), the time densities derived for the neighbouring uncalibrated nodes (blue densities), and the estimated marginal density for the calibrated node (purple)

If a major conflict is observed between a calibration and a marginal density (which we do not observe in our analyses as shown in Supplementary Figure 6, nor was observed in the analyses criticised by Budd and Mann), then it is important to revise the calibration densities used and adjust them to better reflect the knowledge of the species and the relevant fossil record – common

practice in timetree inference analyses and widely emphasised in the literature (e.g., dos Reis et al 2015; dos Reis, Donoghue, Yang 2016; Nascimento et al 2017).

Cited papers:

dos Reis et al 2015: <https://doi.org/10.1016/j.cub.2015.09.066>

dos Reis, Donoghue, Yang 2016: <https://doi.org/10.1038/nrg.2015.8>

Nascimento et al. 2017: <https://doi.org/10.1038/s41559-017-0280-x>

Minor comments:

L82-85: Sentence does not read well. Revise.

We have now revised this sentence for clarity, writing:

"Dating the root of a tree is difficult because errors propagate from the tips to the root of the dated phylogeny and information is not available to estimate the rate of evolution for the branch incident on the root node. Therefore, we analysed genes that duplicated before LUCA with two (or more) copies in LUCA's genome¹⁷. The root in these gene trees represents this duplication preceding LUCA, whereas LUCA is represented by two descendent nodes. Use of these universal paralogues also has the advantage that the same calibrations can be applied at least twice. After duplication, the same species divergences are represented on both sides of the gene tree^{18,19} thus, can be assumed to have the same age. This considerably reduces the uncertainty when genetic distance (branch length) is resolved into absolute time and rate. When a shared node is assigned a fossil calibration, such cross-bracing also serves to double the number of calibrations on the phylogeny improving divergence time estimates. "

L102-106: Why is the dataset limited to these five paralog pairs? Are there not more paralogs that predate LUCA?

We have now addressed this in response to the major comment above.

L158-160: Reads awkwardly.

We have revised these sentences in line with the comments also made by the other reviewer writing the following:

"By using modern prokaryotic genomes as training data, we used a predictive model to estimate the genome size, and the number of protein families LUCA encoded, based on the relationship between the number of KEGG gene families and total number of proteins encoded by modern prokaryote genomes (Extended Data Fig. 3 and 4). Our predictive model estimates a genome size of 2.75Mb (2.49Mb - 2.99Mb) encoding 2657 (2451 - 2855) proteins (see Methods)."

How do the inferred probabilities change with taxon sample size? It would be interesting to see how robust the methods are to this.

We agree that this is an interesting question, in order to investigate this, we performed a analysis where we subsampled the species and gene-trees to 10% and performed reconciliation analysis on these subsampled data sets (four separate tests in total, two replicates with Asgard archaea and two without sampling from Asgard archaea). From these analyses, we found an increase in the overall number of KEGG gene families predicted to be in LUCA; the summed probabilities range from 3000 (without the inclusion of asgards) to 3269 (with asgards). One potential explanation for this, is that the method is conservative in the face of gene tree error: the more taxa, the less likely a given gene tree will match the species topology, and this subsequently results in a lower number of inferred genes in LUCA.

Check the spelling of Wood-Ljungdahl throughout the text.

Thanks, we have now fixed the typo.

L275: Please provide some examples of these alternative electron donors.

We now provide a list of alternative electron donors with the revised sentence: *'Modern acetogens can grow autotrophically on H₂ (and CO₂) or heterotrophically on a wide range of alternative electron donors including: alcohols, sugars and carboxylic acids.'*

L239: Explain what the function of the ffh gene is.

We now explain the gene functions, writing:

"Compared to previous estimates of LUCA's gene content, we find 81 overlapping COG gene families with the consensus data set of Crapitto *et al.*⁸ and 69 overlapping KOs with the dataset of Weiss *et al.*². Key points of agreement between previous studies include the presence of signal recognition particle protein, ffh (COG0541/K03106)⁸ used in the targeting and delivery of proteins for the plasma membrane, a high number of aminoacyl-tRNA synthetases for amino-acid synthesis, and glycolysis/gluconeogenesis enzymes. "

L361: Change to origin of life.

Done.

Fig. 1: Depict/annotate fossil calibrations used.

We have updated Figure 1 accordingly.

Decision Letter, first revision:

25th April 2024

Dear Ed,

Your revised manuscript entitled "The nature of the Last Universal Common Ancestor and its impact on the early Earth system" has now been seen by the same reviewers, whose comments are attached. The reviewers believe the manuscript has improved in revision but the first reviewer has two concerns which will need to be addressed before we can offer publication in Nature Ecology & Evolution. We will therefore need to see your responses to the criticisms raised before we can reach a final decision regarding publication.

We therefore invite you to revise your manuscript taking into account all reviewer and editor comments. Please highlight all changes in the manuscript text file in Microsoft Word format.

* If you have not done so already please begin to revise your manuscript so that it conforms to our Article format instructions at <http://www.nature.com/natecolevol/info/final-submission>. Refer also to any guidelines provided in this letter.

[REDACTED]

Note: This URL links to your confidential home page and associated information about manuscripts you may have submitted, or that you are reviewing for us. If you wish to forward this email to co-

16authors, please delete the link to your homepage.

Nature Ecology & Evolution is committed to improving transparency in authorship. As part of our efforts in this direction, we are now requesting that all authors identified as 'corresponding author' on published papers create and link their Open Researcher and Contributor Identifier (ORCID) with their account on the Manuscript Tracking System (MTS), prior to acceptance. ORCID helps the scientific community achieve unambiguous attribution of all scholarly contributions. You can create and link your ORCID from the home page of the MTS by clicking on 'Modify my Springer Nature account'. For more information please visit www.springernature.com/orcid.

[REDACTED]

Reviewers' comments:

Reviewer #1 (Remarks to the Author):

The authors have thoroughly and effectively addressed nearly all of my comments in this revised manuscript. However, my initial concern over the LUCA genome/proteome size estimates remains and, now that I have a better understanding of the method, I also have a new concern.

Based on the new Methods text, this estimate appears to have been reached by first running a Monte Carlo simulation to estimate the number of LUCA protein families based on their PP values, then running a regression analysis based on extant prokaryotic genomes in order to estimate protein number and genome size as a function of the number of protein families. I believe my original concern, that the initial estimate of the number of protein families is too sensitive to the number of protein families being considered and the accuracy of lower range probability scores, still stands. Based on the supplemental data, the number of protein families inferred to be present in the LUCA based on this estimate (1298 families) would have to include over 200 protein families with PPs < 0.2 (or even more protein families at a lower range of PP values). As I mentioned in the first review, the approach is mathematically correct given the probability estimates. But this still seems like a lot of protein families with low PP values.

My new concern is that the relationship between the number of protein families and the number of proteins is based on an untested assumption that the number of paralogs in the LUCA genome is

17similar to the number of paralogs in modern prokaryotic genomes. This would imply that hundreds of paralogs existed in the LUCA genome, while we only know of about 10-15 universal paralogs in the literature. The new data in Supplemental Figure 4 suggests that the rate of duplications in the LUCA is similar to the rate across all nodes, but do these similar rates indicate that the total number of paralogs in the LUCA genome should be similar to the number of paralogs in extant genomes? Can gene loss in deep nodes account for enough post-LUCA loss of paralogs that we are now left with only a handful of universal paralogs when the actual LUCA genome had hundreds? I would be very interested to know the answers to these questions and I note that the authors do seem to have the relevant data.

I don't believe that either of these comments necessitate a major revision and I also do not want to hold up publication of the manuscript. But the estimated LUCA genome/proteome size is outside of the normal range of LUCA proteome estimates, requires an inclusion of many protein families with very low PPs, and requires a large number of paralogs in the LUCA proteome that are no longer universally conserved. I would strongly recommend that these issues be discussed in greater detail in the final version of the manuscript.

Other comment:

In the supplemental data files, the file extensions are *.csv, but the files are actually in *.tsv format.

Reviewer #2 (Remarks to the Author):

All my concerns have been appropriately addressed by the authors. Thank you for your thorough responses.

*****END*****

Author Rebuttal, first revision:Reviewer #1 (Remarks to the Author):

The authors have thoroughly and effectively addressed nearly all of my comments in this revised manuscript.

Response: We thank the reviewer for acknowledging our efforts to address their comments. We greatly appreciate the feedback and their valuable input which we believe has strengthened the quality of our manuscript.

However, my initial concern over the LUCA genome/proteome size estimates remains and, now that I have a better understanding of the method, I also have a new concern. Based on the new Methods text, this estimate appears to have been reached by first running a Monte Carlo simulation to estimate the number of LUCA protein families based on their PP values, then running a regression analysis based on extant prokaryotic genomes in order to estimate protein number and genome size as a function of the number of protein families. I believe my original concern, that the initial estimate of the number of protein families is too sensitive to the number of protein families being considered and the accuracy of lower range probability scores, still stands. Based on the supplemental data, the number of protein families inferred to be present in the LUCA based on this estimate (1298 families) would have to include over 200 protein families with PPs < 0.2 (or even more protein families at a lower range of PP values).

As I mentioned in the first review, the approach is mathematically correct given the probability estimates. But this still seems like a lot of protein families with low PP values.

Response: Thank you for this comment. We agree that the approach is mathematically correct. There is potentially a conceptual issue here, that also emerges when the reviewer refers to our estimate being “outside the normal range of LUCA proteome estimates” (below). When estimating LUCA’s proteome, there are two distinct goals: the first is to identify a set of gene families which can individually be predicted to have been present in LUCA with high confidence, while the second is to obtain an estimate of its gene content and genome size.

The first goal is the one most common in the literature, and it tends to yield fairly small gene repertoires on the order of several hundred families, enriched for ribosomal proteins and other core genes. This is because, given HGT, differential gene loss and the combination of limited sequence lengths and the long timescales involved make it very difficult to establish with confidence that a particular family was in LUCA. Notably, the converse is not generally true: it is possible to establish for most families with high confidence that they were not present in LUCA. The second aim, reconstructing LUCA’s proteome to obtain an estimate of its gene content and genome size, is a different task from the first that requires a different approach, because an unbiased estimate of genome size necessitates averaging over the uncertainty in gene presence and absence.

20In our study, we do both of these things: we estimate the probability that each gene was present in LUCA, and then we use those probabilities either to define a high-confidence subset of genes that were likely to be in LUCA (e.g., the 399 gene families we identify with PP > 0.75 and that are present in at least some Archaea and Bacteria); this set is directly comparable to, for example, the small core of genes identified in e.g. Weiss et al. (2016) and other previous studies. But then we also attempt to estimate LUCA's genome size using all of the families and their associated probabilities. Any one "draw" of gene content from these probabilities by necessity contains many families with low PPs, because the presence/absence signal for LUCA is quite diffuse. However, an estimate of genome size based on only high-confidence families would be systematically and significantly biased downward, because there is a large pool of low-probability families which likely have contributed at least some genes to LUCA's genome.

It is important to note that our metabolic reconstruction of LUCA does not rely on these low-probability families, but rather on families with high presence probabilities. To make the statistical situation clearer, we have now more explicitly unpacked this setup in the main text, writing:

"Based on the PPs for KEGG KO gene families, we identified a conservative subset of 399 KOs that were likely to be present in LUCA, with PPs ≥ 0.75 and found in both Archaea and Bacteria (Table S1); these families form the basis of our metabolic reconstruction. However, by integrating over the inferred PPs of all KO gene families, including those with low probabilities, we also estimate LUCA's genome size"

To investigate the sensitivity of the estimate, we plotted the effect of discarding (KO) gene families across different thresholds (Figure 1). For example, when discarding 80% of the gene families with the lowest PP values, there are still roughly 1000 gene families inferred as being present, when taking the summed PPs of the non-discarded gene families (See Fig. 1 top). Therefore, the estimate is not unduly affected by genes with low PPs.

My new concern is that the relationship between the number of protein families and the number of proteins is based on an untested assumption that the number of paralogs in the LUCA genome is similar to the number of paralogs in modern prokaryotic genomes. This would imply that hundreds of paralogs existed in the LUCA genome, while we only know of about 10-15 universal paralogs in the literature.

The new data in Supplemental Figure 4 suggests that the rate of duplications in the LUCA is similar to the rate across all nodes, but do these similar rates indicate that the total number of paralogs in the LUCA genome should be similar to the number of paralogs in extant genomes? Can gene loss in deep nodes account for enough post-LUCA loss of paralogs that we are now left with only a handful of universal paralogs when the actual LUCA genome had hundreds? I would be very interested to know the answers to these questions and I note that the authors do seem to have the relevant data.

Response: First, we do not find it surprising that our analyses support the view that LUCA had a similar number of paralogues to modern prokaryotes. It is certainly true that there are very few LUCA paralogues that have evolved vertically, without loss or transfer, to the present day: this number boils down to the handful of paralogues we analyse in the dating part of our study. But note that there are very few genes of any kind that have evolved vertically since the time of LUCA. Our analysis also includes paralogues that have experienced loss, transfer, and additional duplications since the time of LUCA - the great majority of ancient paralogues. For context, it is also worth noting that, while the analysis supports similar numbers of paralogues for LUCA and modern prokaryotes, the overall number of paralogues is low in both cases, with an average of 0.16 gene copies per KO gene family in modern genomes and 0.18 in LUCA.

Second, and of direct relevance for our genome size estimate, we note that our method for inferring genome size does not count paralogues, but simply looks at KO gene family membership: that is, is a genome represented in a particular KO gene family, or not? This makes the inference robust to the number of inferred paralogues per gene family. The relationship between KO gene family membership and genome size is relatively tightly constrained today, especially at the lower-end (see Extended Data Figure 4).

While we therefore do not agree that paralogues cause a specific issue for our analysis, we do appreciate the more general point that our inferences (both of genome size, and of genome composition) are driven by the assumptions of the reconciliation model. While we did acknowledge this in the earlier version of the manuscript, we've now made it clearer still in this part of the methods, writing:

Figure 1. The gene content of LUCA across different probabilities. Top: percentage of lowest support discarded. Middle: across different threshold PPs, (log-scale). Bottom: across different threshold PPs. For each graph, the Y-axis is the number of gene families (calculated through summing the PPs across the included gene families).

“ To ensure that our inference of genome size is robust to uncertainty in the number of paralogues that can be expected to have been present in LUCA. We are using the presence of probability for each of these KEGG KO gene families rather than the estimated copy number.”

I don't believe that either of these comments necessitate a major revision and I also do not want to hold up publication of the manuscript.

Response: We appreciate the reviewer has specified that these comments do not necessitate a major revision, nor hold up publication.

But the estimated LUCA genome/proteome size is outside of the normal range of LUCA proteome estimates, requires an inclusion of many protein families with very low PPs, and requires a large number of paralogs in the LUCA proteome that are no longer universally conserved. I would strongly recommend that these issues be discussed in greater detail in the final version of the manuscript.

Response: This is now addressed in our responses (above) and our updated manuscript.

Other comment:

*In the supplemental data files, the file extensions are *.csv, but the files are actually in *.tsv format.*

Response: Thanks for spotting this, we have now updated the file extension accordingly.

Decision Letter, second revision:

9th May 2024

Dear Ed,

Thank you for submitting your revised manuscript "The nature of the Last Universal Common Ancestor and its impact on the early Earth system" (NATECOLEVOL-23122992B). It has now been seen again by the original reviewers and their comments are below. The reviewers find that the paper has improved in revision, and therefore we'll be happy in principle to publish it in Nature Ecology & Evolution, pending minor revisions to comply with our editorial and formatting guidelines.

[REDACTED]

Reviewer #1 (Remarks to the Author):

The revised manuscript addresses my final concern. Overall, this is an excellent article and I look forward to its publication.

Our ref: NATECOLEVOL-23122992B

21st May 2024

Dear Dr. Moody,

Thank you for your patience as we've prepared the guidelines for final submission of your Nature Ecology & Evolution manuscript, "The nature of the Last Universal Common Ancestor and its impact on the early Earth system" (NATECOLEVOL-23122992B). Please carefully follow the step-by-step instructions provided in the attached file, and add a response in each row of the table to indicate the

24changes that you have made. Please also check and comment on any additional marked-up edits we have proposed within the text. Ensuring that each point is addressed will help to ensure that your revised manuscript can be swiftly handed over to our production team.

****We would like to start working on your revised paper, with all of the requested files and forms, as soon as possible (preferably within two weeks). Please get in contact with us immediately if you anticipate it taking more than two weeks to submit these revised files.****

In recognition of the time and expertise our reviewers provide to Nature Ecology & Evolution's editorial process, we would like to formally acknowledge their contribution to the external peer review of your manuscript entitled "The nature of the Last Universal Common Ancestor and its impact on the early Earth system". For those reviewers who give their assent, we will be publishing their names alongside the published article.

Nature Ecology & Evolution offers a Transparent Peer Review option for new original research manuscripts submitted after December 1st, 2019. As part of this initiative, we encourage our authors to support increased transparency into the peer review process by agreeing to have the reviewer comments, author rebuttal letters, and editorial decision letters published as a Supplementary item. When you submit your final files please clearly state in your cover letter whether or not you would like to participate in this initiative. Please note that failure to state your preference will result in delays in accepting your manuscript for publication.

Cover suggestions

We welcome submissions of artwork for consideration for our cover. For more information, please see our guide for cover artwork.

Nature Ecology & Evolution has now transitioned to a unified Rights Collection system which will allow our Author Services team to quickly and easily collect the rights and permissions required to publish your work. Approximately 10 days after your paper is formally accepted, you will receive an email in

25providing you with a link to complete the grant of rights. If your paper is eligible for Open Access, our Author Services team will also be in touch regarding any additional information that may be required to arrange payment for your article.

Please note that *Nature Ecology & Evolution* is a Transformative Journal (TJ). Authors may publish their research with us through the traditional subscription access route or make their paper immediately open access through payment of an article-processing charge (APC). Authors will not be required to make a final decision about access to their article until it has been accepted. Find out more about Transformative Journals

Authors may need to take specific actions to achieve compliance with funder and institutional open access mandates. If your research is supported by a funder that requires immediate open access (e.g. according to Plan S principles) then you should select the gold OA route, and we will direct you to the compliant route where possible. For authors selecting the subscription publication route, the journal's standard licensing terms will need to be accepted, including [a href="https://www.nature.com/nature-portfolio/editorial-policies/self-archiving-and-license-to-publish"](https://www.nature.com/nature-portfolio/editorial-policies/self-archiving-and-license-to-publish). Those licensing terms will supersede any other terms that the author or any third party may assert apply to any version of the manuscript.

Please use the following link for uploading these materials:
[REDACTED]

[REDACTED]

Reviewer #1:

Remarks to the Author:

The revised manuscript addresses my final concern. Overall, this is an excellent article and I look forward to its publication.

Final Decision Letter:

264th June 2024

Dear Ed,

We are pleased to inform you that your Article entitled "The nature of the Last Universal Common Ancestor and its impact on the early Earth system", has now been accepted for publication in *Nature Ecology & Evolution*.

Over the next few weeks, your paper will be copyedited to ensure that it conforms to *Nature Ecology and Evolution* style. Once your paper is typeset, you will receive an email with a link to choose the appropriate publishing options for your paper and our Author Services team will be in touch regarding any additional information that may be required

Due to the importance of these deadlines, we ask you please us know now whether you will be difficult to contact over the next month. If this is the case, we ask you provide us with the contact information (email, phone and fax) of someone who will be able to check the proofs on your behalf, and who will be available to address any last-minute problems . Once your paper has been scheduled for online publication, the Nature press office will be in touch to confirm the details.

Acceptance of your manuscript is conditional on all authors' agreement with our publication policies (see www.nature.com/authors/policies/index.html). In particular your manuscript must not be published elsewhere and there must be no announcement of the work to any media outlet until the publication date (the day on which it is uploaded onto our web site).

Please note that *Nature Ecology & Evolution* is a Transformative Journal (TJ). Authors may publish their research with us through the traditional subscription access route or make their paper immediately open access through payment of an article-processing charge (APC). Authors will not be required to make a final decision about access to their article until it has been accepted. Find out more about Transformative Journals

Authors may need to take specific actions to achieve compliance with funder and institutional open access mandates. If your research is supported by a funder that requires immediate open access (e.g. according to Plan S principles) then you should select the gold OA route, and we will direct you to the compliant route where possible. For authors selecting the subscription publication route, the journal's standard licensing terms will need to be accepted, including [a href="https://www.nature.com/nature-portfolio/editorial-policies/self-archiving-and-license-to-publish"](https://www.nature.com/nature-portfolio/editorial-policies/self-archiving-and-license-to-publish). Those licensing terms will supersede any other terms that the author or any third party may assert apply to any version of the manuscript.

In approximately 10 business days you will receive an email with a link to choose the appropriate publishing options for your paper and our Author Services team will be in touch regarding any

27additional information that may be required.

We welcome the submission of potential cover material (including a short caption of around 40 words) related to your manuscript; suggestions should be sent to Nature Ecology & Evolution as electronic files (the image should be 300 dpi at 210 x 297 mm in either TIFF or JPEG format). Please note that such pictures should be selected more for their aesthetic appeal than for their scientific content, and that colour images work better than black and white or grayscale images. Please do not try to design a cover with the Nature Ecology & Evolution logo etc., and please do not submit composites of images related to your work. I am sure you will understand that we cannot make any promise as to whether any of your suggestions might be selected for the cover of the journal.

You can generate the link yourself when you receive your article DOI by entering it here: <http://authors.springernature.com/share>.

[REDACTED]

P.S. Click on the following link if you would like to recommend Nature Ecology & Evolution to your librarian <http://www.nature.com/subscriptions/recommend.html#forms>

** Visit the Springer Nature Editorial and Publishing website at www.springernature.com/editorial-and-publishing-jobs for more information about our career opportunities. If you have any questions please click here.**